# Gallic Acid Alleviates Acetaminophen-Induced Acute Liver Injury by Regulating Inflammatory and Oxidative Stress Signaling Proteins

**DOI:** 10.3390/antiox14070860

**Published:** 2025-07-14

**Authors:** Jing Zhao, Yuan Zhao, Shuzhe Song, Sai Zhang, Guodong Yang, Yan Qiu, Weishun Tian

**Affiliations:** 1College of Animal Science and Technology, Henan University of Science and Technology, Luoyang 471000, China; hkdzhao1230@haust.edu.cn (J.Z.); 230320181082@stu.haust.edu.cn (S.S.); 220320181469@stu.haust.edu.cn (S.Z.); 9905001@haust.edu.cn (G.Y.); dkqiuyan@haust.edu.cn (Y.Q.); 2Institute of Traditional Chinese Veterinary Medicine, College of Veterinary Medicine, Nanjing Agricultural University, Nanjing 210095, China; zhaoy@stu.njau.edu.cn

**Keywords:** gallic acid, APAP, acute liver injury, oxidative stress, inflammation

## Abstract

Acetaminophen (APAP) overdose is a major cause of drug-induced liver injury (DILI) globally, which necessitates effective therapies. Gallic acid (GA), a naturally abundant polyphenol, possesses potent antioxidant and anti-inflammatory properties that may overcome the limitations of N-acetylcysteine (NAC), such as its narrow therapeutic window. This study systematically investigated the hepatoprotective effects and underlying molecular mechanisms of GA against APAP-induced acute liver injury (ALI). Mice received an intraperitoneal injection of APAP (300 mg/kg), followed by an oral administration of GA (50 or 100 mg/kg) or NAC (150 mg/kg) 1 h post-intoxication. Both GA and NAC significantly ameliorated hypertrophy and histopathological damage, as evidenced by reduced serum ALT/AST levels and inflammatory cytokines. TUNEL staining revealed a marked suppression of apoptotic and necrotic cell death, further supported by the downregulation of pro-apoptotic Bax and the upregulation of anti-apoptotic Bcl-2 mRNA expression. GA and NAC treatment restored hepatic glutathione (GSH) content, enhanced antioxidant enzyme gene expression, and reduced malondialdehyde (MDA) accumulation. Mechanistically, GA and NAC inhibited MAPK phosphorylation while activating AMPK signaling. Taken together, these findings demonstrate that GA mitigates APAP-induced ALI by modulating oxidative stress and inflammation through the regulation of MAPK/AMPK signaling proteins.

## 1. Introduction

Acetaminophen (APAP) is a classic antipyretic and analgesic used to treat fever and headache associated with cold or influenza. At the therapeutic dosage, over 90% APAP undergoes hepatic conjugation via uridine diphosphate glucuronosyltransferase (UGT) and sulfotransferases, forming nontoxic glucuronide and sulfate metabolites excreted via biliary and renal pathways [1]. However, APAP administration overdose disrupts hepatic metabolic homeostasis, leading to acute liver injury (ALI), acute liver failure (ALF), and even death [2,3]. APAP-induced hepatotoxicity is becoming an important issue in the development of clinical ALF in many countries, with approximately 400,000 deaths each year in China [4,5]. Acetaminophen (APAP) hepatotoxicity primarily stems from the cytochrome P450-mediated formation of N-acetyl-p-benzoquinone imine (NAPQI), which depletes glutathione (GSH) reserves and triggers mitochondrial oxidative stress, culminating in necrotic cell death and sterile inflammation [6,7,8,9]. Consequently, elucidating the molecular pathogenesis of APAP hepatotoxicity is critical for developing targeted therapeutics to mitigate the rising global burden of ALF.

In APAP-induced ALI, molecular pathogenesis involves reactive oxygen species (ROS)-driven sustained JNK activation. Phosphorylated JNK translocates to mitochondria, binding Sab to inhibit electron transport chain complexes, and promotes the opening of the mitochondrial permeability transition pore, thereby worsening the bioenergetic crisis and apoptosis [7,10,11,12]. Pharmacological JNK/ERK inhibition (e.g., SP600125/PD98059) attenuates hepatocellular damage by reducing ROS generation, suppressing caspase-3 cleavage, and preserving ATP synthesis, underscoring MAPK’s centrality in APAP toxicity [7,12,13]. Concurrently, the AMP-activated protein kinase (AMPK) serves as a metabolic rheostat coordinating energy homeostasis and redox balance. AMPK activation enhances cellular antioxidant capacity to resist oxidative stress via Nrf2/HO-1 signaling pathways [14,15]. Previous studies revealed that GA activates the Keap1-Nrf2-ARE and NF-κB pathway to regulate oxidative stress and inflammation [16,17]. Moreover, GA also influences kinases like PI3K, Akt, MAPK, and AMPK to attenuate LPS-induced inflammation and oxidative stress [18].

The current clinical treatment for APAP-induced liver injury relies on N-acetylcysteine (NAC). However, its narrow therapeutic window and adverse effects (including nausea and anaphylaxis) limit clinical utility [19,20,21,22]. To address these limitations, identifying alternative hepatoprotective agents is imperative. Gallic acid (GA) is a natural phenolic compound with established antioxidant properties; it demonstrates protective effects against oxidative damage, anti-inflammatory activity, and hepatoprotective potential [23,24]. But the therapeutic potential and molecular mechanisms of GA in APAP-induced liver injury remain uncharacterized. Therefore, this study aims to clarify these aspects through biochemical, histological, and molecular analyses, including assessments of superoxide production, oxidative stress markers, inflammation, and apoptosis, and further define the protective effects and underlying pathways of GA in APAP-induced ALI.

## 2. Materials and Methods

### 2.1. Animal Housing

All animal experiments were conducted in accordance with protocols reviewed and approved by the Laboratory Animal Center of Henan University of Science and Technology (Approval No. HAUST-024-M0924011). Male C57BL/6 mice (6 weeks, 25 ± 2 g) were housed at an environmental temperature of 25 ± 1 °C and a humidity of 55% ± 10%, with a light/dark cycle of 12 h. All were ensured free access to a diet.

### 2.2. Experimental Design

Following a 7-day acclimation period, mice were randomly divided into 5 groups (n = 8 per group). To evaluate the protective effects of GA against APAP-induced ALI, mice received an intraperitoneal injection of APAP (300 mg/kg), except for those in the control group. Mice received an intragastric administration of GA (50 or 100 mg/kg) or N-acetylcysteine (NAC; 150 mg/kg) 1 h post-APAP injection. GA and NAC were separately dissolved in saline prior to oral administration. The dosages for APAP, GA, and NAC were selected according to previous experiments [24,25,26]. Control mice received an equivalent volume of saline instead of APAP and subsequent treatments. Based on established protocols, mice were anesthetized 12 h after the intraperitoneal injection of APAP. Blood samples were then collected via cardiac puncture. Subsequently, mice were sacrificed by cervical dislocation, and liver tissues were harvested immediately [25,27]. Blood samples and liver tissues were collected for biochemical, histological, and molecular analysis.

### 2.3. Liver Injury Markers Analyses

Blood was centrifuged to obtain serum samples (3000 g, 15 min, and 4 °C). The colorimetric method was used to detect alanine aminotransferase (ALT) and aspartate aminotransferase (AST) via commercial assay kits (ASAN Pharmaceutical, Hwasung, South Korea), and the pathogenesis of inflammation marker NO was detected by the Griess assay kit (Thermo Fisher Scientific, Waltham, MA, USA). Optical density was measured at 490 nm (ALT, AST) and 548 nm (NO) absorbance by a tunable Versa Max microplate reader (Molecular Devices, Sunnyvale, CA, USA). Three independent experiments were performed for statistical analysis.

### 2.4. Liver Histological Investigation

Mouse liver was fixed in 4% paraformaldehyde (PFA) in a phosphate-buffered saline (PBS) solution and embedded in paraffin wax. Liver tissues were sliced at 4 µm thickness and stained with hematoxylin and eosin (H&E). Digital images were captured under a light microscope (Leica Microsystems, Wetzlar, Germany) at magnifications of 100× and 400×. Apoptotic and necrotic cell death were assessed on paraffin-embedded sections by terminal deoxynucleotidyl transferase dUTP nick end labeling (TUNEL) assays using an in situ apoptosis detection kit (Chemicon, Temecula, CA, USA) in accordance with the manufacturer’s directions. The diaminobenzidine (DAB)-positive areas of the liver sections were assessed by Digital image analysis software (analySIS TS, Olympus Corp., Tokyo, Japan).

### 2.5. Oxidative Stress Markers Tests

Liver samples were homogenized and centrifuged. The supernatant was used to analyze hepatic GSH and malondialdehyde (MDA) levels using GHS/GSSH quantification (Dojindo Molecular Technologies, Inc., Kumamoto, Japan) and TBARS kits (Cayman, Ann Arbor, MI, USA) according to the suppliers’ instructions. Optical density was measured at 535 nm (MDA) and 405 nm (GSH) absorbance by a tunable Versa Max microplate reader (Molecular Devices, Sunnyvale, CA, USA). Three independent experiments were performed for statistical analysis.

### 2.6. Hepatic Cytokines Level Assay

Liver tissues were homogenized, and hepatic TNF-α, IL-6, and IL-1β levels were measured using ELISA kits (Thermo Scientific, Waltham, MA, USA) following the manufacturer’s instructions. Optical density was measured at 450 nm absorbance by a tunable Versa Max microplate reader (Molecular Devices, Sunnyvale, CA, USA). Three independent experiments were performed for statistical analysis.

### 2.7. qPCR Analysis

A TRIzol reagent (Invitrogen, Carlsbad, CA, USA) was used to isolate the total hepatocellular RNA from the liver tissue. RNA (3 µg) was used to reverse transcribe complementary DNA (cDNA) using a commercial biochemical kit (Thermo Scientific, Waltham, MA, USA). qPCR was performed using SYBR Green qPCR Kit (TOYOBO, Tokyo, Japan) via the CFX96™ Real-Time PCR Detection System (Bio-Rad Laboratories, Hercules, CA, USA). Prior to experimental use, all primer pairs underwent validation for amplification efficiency and specificity. Amplification efficiency was determined via standard curves generated from five-point ten-fold serial dilutions of cDNA (E = 95–105%; slope = −3.3 to −3.5; R^2^ > 0.99). Melt curve analysis (65–95 °C, 0.5 °C/5 s increments) confirmed single-amplification products with sharp, singular peaks and the absence of primer dimers. The cycling conditions for amplification were 95 °C for 2 min, followed by 40 cycles at 94 °C for 60 s and 60 °C for 90 s. Expression was calculated by comparing the cycle threshold (Ct) values of samples normalized to glyceraldehyde-3-phosphate dehydrogenase (GAPDH) as an internal control. Three independent experiments were performed for statistical analysis. All PCR primers are summarized in Table 1.

### 2.8. Functional Enrichment Analysis of GO and KEGG Pathways

To elucidate the biological implications of potential targets in APAP-induced ALI, Gene Ontology (GO) annotation and Kyoto Encyclopedia of Genes and Genomes (KEGG) pathway analyses were conducted through the DAVID bioinformatics platform (version 6.8, https://david.ncifcrf.gov/ (accessed on: 26 November 2024)). The GO analysis systematically categorized gene functions into three ontologies: biological processes, cellular components, and molecular functions. Concurrently, KEGG enrichment analysis identified significantly perturbed signaling pathways associated with ALI pathogenesis. Enrichment results were visualized using the Weishengxin platform (https://www.bioinformatics.com.cn/ (accessed on: 7 January 2025)).

### 2.9. Molecular Docking Validation

The structure-based validation of network pharmacology-derived targets was conducted using AutoDock Vina (v1.2.0) and PyMOL (v2.5.2). The high-resolution crystal structures (<2.5 Å) of target proteins were obtained from RCSB PDB (https://www.rcsb.org/ (accessed on: 8 January 2025)), while the 3D conformation of GA (PubChem CID: 370) was retrieved from the PubChem database. Protein structures underwent preprocessing in PyMOL, including the removal of crystallographic water/heteroatoms and the addition of polar hydrogens with Kollman charges. Top-ranked binding poses were selected based on binding energy and structural stability (RMSD < 2.0 Å), followed by the visualization of intermolecular interactions (hydrogen bonds and π–π stacking) using PyMOL’s ligand interaction module.

### 2.10. Western Blotting Assay

The total protein of liver samples was extracted using a lysis buffer with a tissue protein extraction reagent (T-PER) (Thermo Scientific, Waltham, MA, USA). Equal amounts of the extracted protein samples were separated by sodium dodecyl sulfate-polyacrylamide gel electrophoresis (SDS-PAGE) and then transferred onto nitrocellulose membranes. Each membrane was incubated in 5% non-fat milk solution for 2 h at room temperature to block non-specific binding sites. Subsequently, the membranes were incubated with diluted primary antibodies, followed by immersion in an HRP-labeled goat anti-rabbit IgG secondary antibody. Protein blots were visualized using the ECL Western Blotting Analysis System. Protein band expression was quantified with Quantity One-4.6.6 (Bio-Rad Laboratories, Hercules, CA, USA). Target protein expression was presented as relative protein expression. Three independent experiments were performed for statistical analysis.

### 2.11. Statistical Analysis

Statistical data were analyzed using GraphPad Prism 9.4.1 (GraphPad Software, La Jolla, CA, USA). Group differences were evaluated by one-way analysis of variance (ANOVA) followed by Tukey’s multiple-comparison test. Data were expressed as mean ± standard error of the mean (SEM) (n = 8). Statistical significance was defined as *p* < 0.05. To address the specific hypothesis regarding GA and liver protection, this study emphasized APAP-induced injuries (control group vs. APAP) and therapeutic effects (APAP vs. treatment group), while the comparative results between the other groups are not presented.

## 3. Results

### 3.1. GA Treatment Mitigates Liver Burden in APAP-Exposed Mice

The overdose of APAP can cause hepatotoxicity, impose a burden on the liver, and lead to hepatomegaly, which can be verified by macroscopic observations. In this study, mice treated with GA and NAC markedly alleviated the adverse effects of APAP on the liver (Figure 1a). Additionally, these findings were corroborated by liver weight measurements. As shown in Table 2’s data, exposure to APAP caused a significant (*p* < 0.05, *p* < 0.01, or *p* < 0.001) increase in the liver weight and liver/body weight indexes. Mice treated with GA displayed a notable (*p* < 0.05, *p* < 0.01, or *p* < 0.001) decrease in liver weight and liver/body weight indexes. Notably, at a dose of 100 mg/kg, GA showed similar protective effects in reducing the liver weight and liver/body weight indexes compared to NAC-treated mice (Figure 1b).

### 3.2. GA Treatment Attenuated the Liver Injury in APAP-Exposed Mice

To further investigate the effects of GA on APAP-induced hepatotoxicity, hepatic histopathological changes were assessed by H&E staining. In the normal control group, liver tissue exhibited intact architecture, with hepatocytes displaying orderly arrangement, well-defined borders, abundant cytoplasm, and no observable abnormalities. In contrast, APAP administration induced severe hepatic damage, characterized by hepatocyte swelling, nuclear pleomorphism, disorganized hepatic cord structures, neutrophil infiltration, and extensive necrosis. However, treatment with GA or NAC markedly attenuated these pathological alterations, significantly reducing the severity and extent of liver injury (Figure 2a).

ALT, AST, and NO are crucial indicators for evaluating liver injury. Serum was collected, and then, ALT, AST, and NO concentrations were upregulated notably (*p* < 0.05, *p* < 0.01, or *p* < 0.001) after mice were challenged with APAP only. But mice treated with GA exhibited marked reversal with respect to liver injury biomarker upregulation (Figure 2b), indicating that GA alleviated the hepatotoxicity caused by APAP and exhibited similar effects to NAC.

### 3.3. GA Treatment Reduced Hepatic Cell Death in APAP-Induced Mice ALI

Apoptotic and necrotic cell death are cardinal manifestations of APAP-induced hepatotoxicity. TUNEL assays were performed to evaluate the effect of GA on APAP-induced hepatocyte apoptosis. Liver sections from mice subjected to 12 h APAP challenge exhibited extensive TUNEL-positive signals, manifested as abundant dark brown granular deposits and apoptotic and necrotic cells concentrated predominantly in the pericentral regions. In contrast, treatment with GA or NAC markedly (*p* < 0.001) ameliorated hepatocyte death, as evidenced by substantially reduced TUNEL staining intensity and distribution (Figure 3a,b).

The anti-apoptotic Bcl-2 and pro-apoptotic Bax genes play pivotal roles in regulating necrotic cell death [28]. qPCR analysis revealed that GA has regulatory effects on necrotic cell-death-related proteins. That is, GA has a positive regulatory effect on the mRNA expressions of anti-apoptotic Bcl-2 and a negative regulatory effect on mRNA expressions of pro-apoptotic BAX (Figure 3c), aligning with the protective effects observed in TUNEL staining.

### 3.4. GA Treatment Attenuated APAP-Induced Hepatic Oxidative Stress

Oxidative stress is a fundamental contributor to APAP-induced hepatotoxicity [19]. MDA, a well-established marker of oxidative stress, was used to quantify lipid peroxidation [29]. GSH, a key primary cellular antioxidant, was measured to assess oxidative damage. Twelve hours after intraperitoneal APAP injection, hepatic MDA levels increased significantly (*p* < 0.001), accompanied by a notable (*p* < 0.05) decrease in GSH levels. GA significantly (*p* < 0.01 or *p* < 0.001) reversed the APAP-induced elevation in MDA levels and prevented GSH depletion (Figure 4a,b). Furthermore, GSH can be excreted from the body in a non-toxic conjugate with NAPQI, a metabolite of APAP, which is formed by the hepatic cytochrome P450 system, particularly CYP2E1. GA treatment markedly (*p* < 0.05) diminished CYP2E1transcription, which resulted in lower CYP2E1 mRNA expression (Figure 4c). Moreover, the mRNA expressions of the antioxidant enzymes catalase, SOD1, and SOD2 were dramatically (*p* < 0.05, *p* < 0.01, or *p* < 0.001) decreased in mice treated with APAP alone. However, GA restored the expression of key antioxidant enzymes suppressed by APAP (Figure 4d–f), further demonstrating its ability to mitigate oxidative stress.

### 3.5. GA Suppresses APAP-Triggered Hepatic Inflammation

The inflammatory response plays a critical role in the pathogenesis of APAP-induced ALI. The ELISA analysis demonstrated that the APAP challenge significantly (*p* < 0.01, or *p* < 0.001) elevated the hepatic levels of pro-inflammatory cytokines (TNF-α, IL-6, and IL-1β), while GA treatment dramatically (*p* < 0.05, *p* < 0.01, or *p* < 0.001) attenuated these increases (Figure 5a–c). The anti-inflammatory effects were further supported at the transcriptional level. The qPCR results demonstrated that APAP exposure markedly (*p* < 0.05, *p* < 0.001) upregulated key inflammatory mediators (COX-2, CYR61, and iNOS), whereas GA significantly (*p* < 0.05, *p* < 0.01, and *p* < 0.001) suppressed their mRNA expression (Figure 5d–f). These findings indicated that GA exerts potent anti-inflammatory effects by modulating both cytokine release and inflammatory signaling pathways.

### 3.6. Integrated Pathway Analysis Reveals Multi-Modal Mechanisms of GA Against APAP-Induced Hepatotoxicity

To delineate the therapeutic mechanisms of GA in APAP-induced liver injury, systematic pathway enrichment analysis was conducted using the DAVID database with shared drug–disease targets. GO enrichment demonstrated tripartite functional engagement. Biological processes prominently featured signal transduction, phosphorylation, inflammatory regulation, and apoptosis. Cellular component mapping revealed the cytosol, cytoplasm, nucleus, and membranes as primary intervention sites. Molecular functions highlighted protein binding and kinase activity, indicating direct molecular targeting (Figure 6a). KEGG pathway analysis identified that GA coordinated the modulation of pathological axes, including NF-κB-mediated inflammation through membrane receptor interactions and MAPK/AMPK-driven redox homeostasis in cytoplasmic compartments (Figure 6b).

### 3.7. Molecular Docking Simulations of GA Interactions with Key Signaling Mediators

Molecular docking is used to explore potential interactions between GA and key signaling mediators. While acknowledging inherent computational limitations, this approach provides predictive insights for identifying mechanistic targets. The calculated free energies (kcal/mol) revealed strong interactions with p38 MAPK (−6.3), JNK (−5.4), ERK (−5.6), AMPK (−5.8), and NF-κB (−3.28), all exceeding the −1.2 kcal/mol threshold for biological relevance (Figure 7, Table 3). Molecular docking predicts the potential binding of GA to JNK/ERK/p38/AMPK/NF-κB, and it is expected to be a mechanism for resolving GA in alleviating liver injury.

### 3.8. GA Treatment Reduced Phosphorylation of NF-κB and MAPKs

To delve deeper into the key inflammation-related proteins underlying the protective role of GA against APAP-stimulated hepatotoxicity in mice, the protein expressions of NF-κB and MAPKs (JNK, ERK, and p38) were meticulously analyzed. NF-κB signaling is a key regulatory mechanism that triggers the production of proinflammatory cytokines, such as TNF-α, IL-1β, and IL-6 [30]. Moreover, MAPKs are able to transduce signals from a diverse array of extracellular stimuli, including oxidative stress and cytotoxic factors [31]. Their activation has been shown to play important roles in APAP-induced hepatotoxicity [32,33]. As evidenced by the Western blotting results, the protein expressions of NF-κB, JNK, ERK, and p38 were markedly (*p* < 0.001) raised in the livers subjected to APAP alone (Figure 8). In contrast, treatment with GA or NAC led to a significant suppression (*p* < 0.05, *p* < 0.01, or *p* < 0.001) of their activation (Figure 8).

### 3.9. GA Enhanced AMPK Phosphorylation to Mitigate Oxidative Stress

AMPK is a positive regulator for resisting oxidative stress induction that plays a defensive role in the APAP-stimulated ALI model [25]. After the APAP challenge, weak Western blotting band intensities of pAMPKα1 were observed in the mouse livers. However, the application of GA to mice with APAP-induced liver injury dramatically (*p* < 0.05) reversed the decreased protein expression of pAMPKα1 (Figure 9a). Moreover, the use of Compound C (an AMPK inhibitor) notably (*p* < 0.05) inhibited the protein expression of pAMPKα1 in APAP-induced liver injury treated with GA (Figure 9b). The above findings further demonstrate that GA can alleviate APAP-induced oxidative stress in the liver by activating the AMPK signaling pathway.

## 4. Discussion

Drug-induced liver injury (DILI) has become a serious public health problem that cannot be ignored. In China, the annual incidence of DILI is estimated to be at least 23.80 per 100,000, with a discernible upward trend [34]. APAP is a primary contributor to DILI, and numerous studies have underscored the pivotal role of natural medicines in mitigating and preventing this condition [35,36,37]. Gallic acid (GA) is a low-molecular-weight polyphenolic compound that is abundantly distributed in botanical sources [23,38]. It exhibits multifaceted bioactivities, including antimicrobial, anti-inflammatory, antioxidant, antitumor, cardiovascular-protective, and hepatoprotective properties [23,24,39]. This study delved into the mechanisms by which GA ameliorates APAP-induced acute liver injury, revealing that GA mitigates APAP-triggered inflammation and oxidative stress through the downregulation of NF-κB and MAPK proteins and the upregulation of the AMPK signaling cascade.

APAP overdose imposes a profound burden on hepatic physiology, manifesting as increased liver weight, an elevated liver/body mass ratio, hepatotoxicity, and the disruption of normal hepatic histology [25,40]. The results demonstrated that GA reversed APAP-induced hepatomegaly, preliminarily validating its therapeutic potential in ALI. Further corroboration came from the reduced serum levels of hepatotoxicity markers (ALT, AST, and NO) and histopathological improvements, including diminished hepatocyte necrosis and inflammatory cell infiltration.

While the role of apoptosis and necrosis in APAP-induced liver injury remains debated, evidence supports its occurrence during the early injury phase, transitioning to secondary necrosis [41]. The metabolites of APAP cause ROS generation and then active JNK phosphorylation. This initiates mitochondrial dysfunction, releases pro-apoptotic Bcl-2 family members, and activates the caspase cascade, culminating in apoptosis [42]. Consistently, GA attenuated centrilobular necrosis and TUNEL-positive cell areas, corroborated by upregulated Bcl-2 and downregulated Bax mRNA expression, highlighting its anti-apoptotic effects.

During APAP metabolism, the reactive intermediate NAPQI forms mitochondrial protein adducts, inducing oxidative stress and promoting glutathione (GSH) depletion [25]. Both GA and NAC normalized APAP-altered oxidative stress markers by reducing malondialdehyde (MDA) levels, restoring GSH content, and upregulating antioxidant enzymes such as catalase, SOD-1, and SOD-2. These findings highlight GA’s critical role in protecting hepatocytes against oxidative damage. Notably, NAC and GA operate via distinct pharmacological mechanisms: NAC acts as a rapid-onset GSH precursor, while GA functions as a multi-target polyphenol with prolonged effects. Their comparison serves as a therapeutic benchmark rather than implying mechanistic equivalence.

CYP2E1 is a pivotal enzyme in the bioactivation of acetaminophen (APAP), which was downregulated by GA. In this study, although the CYP2E1 protein’s expression was not measured, the observed upregulation of CYP2E1 mRNA aligns with elevated serum ALT/AST levels, increased hepatic malondialdehyde (MDA) content, and histological evidence of centrilobular necrosis. These endpoints collectively reflected CYP2E1-mediated oxidative stress, which is consistent with previous research [43,44]. This downregulation effectively mitigated the accumulation of NAPQI and consequently curbed the subsequent overproduction of reactive oxygen species (ROS) [45,46,47]. Additionally, APAP-induced hepatocyte necrosis releases damage-associated molecular patterns (DAMPs), activating inflammatory receptors and exacerbating liver injury [48,49]. GA suppressed proinflammatory cytokines (TNF-α, IL-1β, and IL-6) and downregulated inflammation-related genes (iNOS, COX-2, and CYR61), interrupting the vicious cycle of sterile inflammation and secondary hepatic damage. GA suppressed proinflammatory cytokines (TNF-α, IL-1β, and IL-6) and downregulated inflammation-related genes (iNOS, COX-2, and CYR61), interrupting the vicious cycle of sterile inflammation and secondary hepatic damage. These collective effects establish GA as a dual-target agent against oxidative stress and inflammatory responses.

Integrative analyses via GO and KEGG databases and molecular docking were performed to investigate the mechanism of GA in APAP-induced ALI. The analysis revealed that GA interacted with the NF-κB, MAPK (JNK, ERK, and p38), and AMPK pathways. Western blotting confirmed that GA increased pAMPKα protein expression while suppressing NF-κB and MAPK phosphorylation. The NF-κB signaling pathway is an extremely important pathway in cell signaling, which plays a key role in regulating various physiological processes such as immune response, inflammatory response, and cell death. NF-κB activation causes the transcription of some pro-inflammatory cytokines, including TNF-α and IL-1β [50]. Our research demonstrates that GA inhibited NF-κB phosphorylation and suppressed the secretion of TNF-α, IL-6, and IL-1β. MAPK signaling acts as a key player in the control of cell growth, differentiation, and apoptosis, as well as stress and immune responses [51]. AMPK activation attenuates oxidative stress by enhancing antioxidant enzyme expression and maintaining redox balance [52,53,54]. In this experimental study, GA inhibited MAPK phosphorylation (ERK, JNK, and p38), thereby suppressing pro-inflammatory transcription, and activated AMPK signaling pathways, thereby upregulating antioxidant genes (catalase, SOD-1, and SOD-2), decreasing MDA levels, and restoring GSH content. Compound C is a well-established selective AMPK inhibitor that is frequently employed in studies of AMPK signaling pathways [15,55]. While primarily functioning through competitive binding to the ATP-binding site of AMPK, it exhibits documented off-target effects that warrant consideration in experimental interpretation. In the current study, the use of Compound C significantly reduced the agonistic effect of GA on AMPK, further confirming that GA can significantly activate the AMPK signaling pathway and thereby regulate the secretion of inflammatory factors and the activity of antioxidant enzymes. This research result is similar to a previous study [55].

While this study elucidates the key hepatoprotective mechanisms of GA against APAP-induced acute liver injury (ALI), several limitations remain. Future investigations will incorporate multi-timepoint analyses, pharmacokinetic profiling via LC-MS/MS, CYP2E1 protein quantification, and sex-stratified models to enhance translational relevance.

## 5. Conclusions

In conclusion, this study demonstrates that GA has protective effects against APAP-induced hepatotoxicity by targeting inflammation and oxidative stress. GA exerts its effects through the coordinated modulation of NF-κB/MAPK and AMPK, thereby attenuating proinflammatory cascades and enhancing antioxidant defense. These findings underscore the therapeutic potential of GA as a natural compound for DILI intervention, providing a framework for further translational research into its clinical application.

## Figures and Tables

**Figure 1 antioxidants-14-00860-f001:**
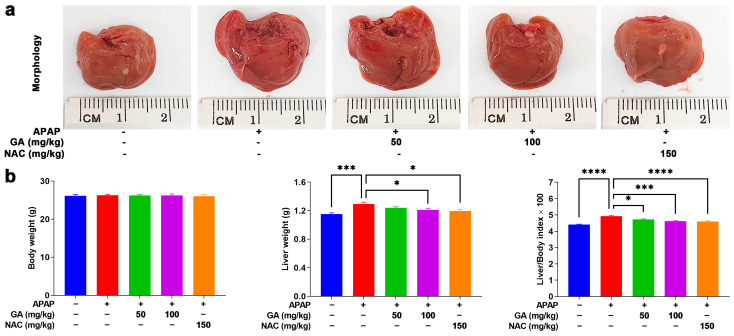
GA attenuates APAP-induced hepatic hypertrophy. (**a**) The livers were harvested following the sacrifice of the mice, and their morphology was recorded. (**b**) The body weights and liver weights of the mice were measured, and their ratio was calculated. The data are presented as the mean ± standard error of the mean (SEM) (n = 8). * *p* < 0.05, *** *p* < 0.001, and **** *p* < 0.0001 denote statistical significance.

**Figure 2 antioxidants-14-00860-f002:**
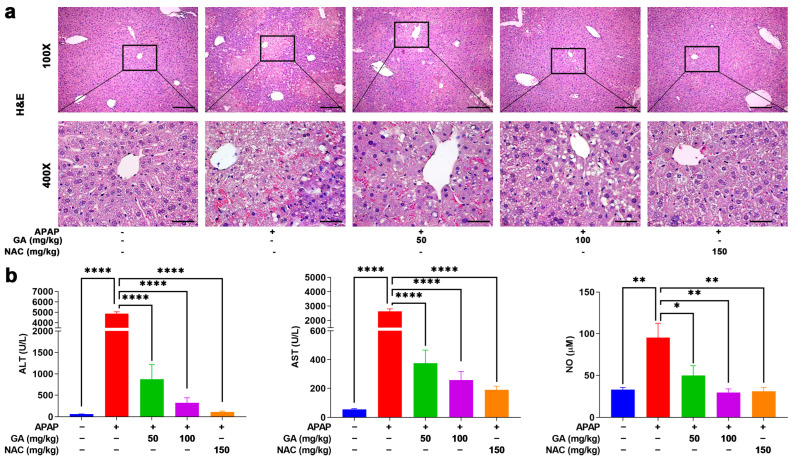
GA mitigates APAP-induced hepatic injury. (**a**) Liver sections were stained with H&E staining. (**b**) Serum ALT, AST, and NO levels were measured. The data are presented as the mean ± standard error of the mean (SEM) (n = 8). * *p* < 0.05, ** *p* < 0.01, and **** *p* < 0.0001 denote statistical significance.

**Figure 3 antioxidants-14-00860-f003:**
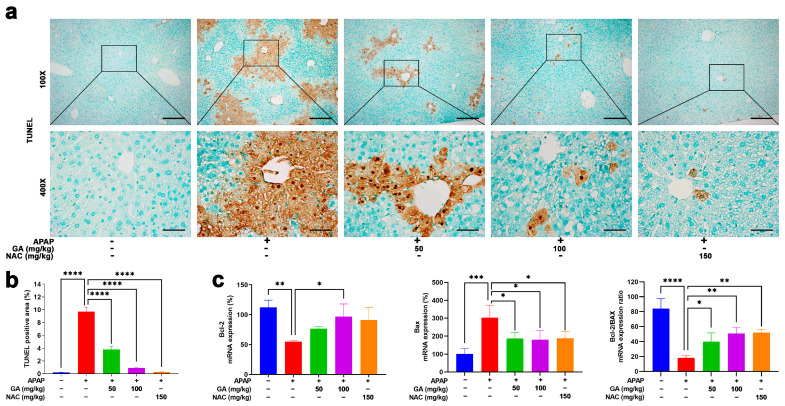
GA mitigates APAP-induced hepatocyte death. (**a**) Representative images of liver sections stained with terminal-deoxynucleotidyl transferase-mediated nick end labeling (TUNEL) are shown. (**b**) Quantitative analysis of the TUNEL-positive area was performed. (**c**) The mRNA expressions of Bcl-2 (anti-apoptotic) and Bax (pro-apoptotic) were evaluated by qPCR, and the ratio of Bcl-2 to Bax was calculated. The data are presented as the mean ± standard error of the mean (SEM) (n = 8). * *p* < 0.05, ** *p* < 0.01, *** *p* < 0.001, and **** *p* < 0.0001 denote statistical significance.

**Figure 4 antioxidants-14-00860-f004:**
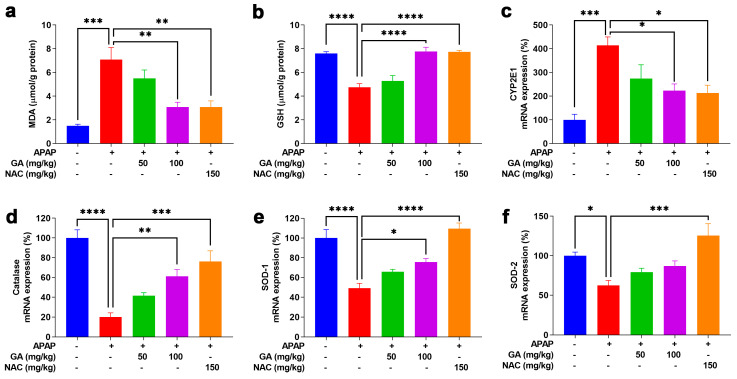
GA attenuated APAP-induced hepatic oxidative stress. (**a**,**b**) Commercial kits were employed to measure the levels of hepatic MDA and glutathione GSH. (**c**) The mRNA expressions of CYP2E1, as well as (**d**–**f**) catalase, SOD-1, and SOD-2, were determined through qPCR analysis. The data are presented as the mean ± standard error of the mean (SEM) (n = 8). * *p* < 0.05, ** *p* < 0.01, *** *p* < 0.001, and **** *p* < 0.0001 denote statistical significance.

**Figure 5 antioxidants-14-00860-f005:**
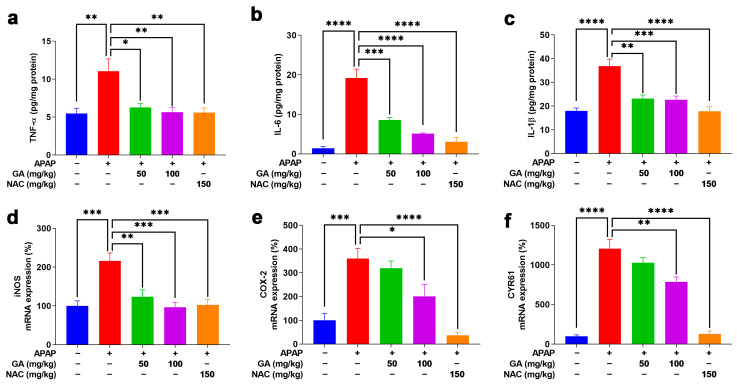
GA attenuated APAP-induced hepatic inflammation. (**a**–**c**) The hepatic pro-inflammatory cytokines, including TNF-α, IL-6, and IL-1β, were quantified using ELISA kits according to the manufacturer’s protocols. (**d**–**f**) The mRNA expression levels of inflammation-related markers, including iNOS, COX-2, and CYR61, were analyzed by qPCR analysis. The data are presented as the mean ± standard error of the mean (SEM) (n = 8). * *p* < 0.05, ** *p* < 0.01, *** *p* < 0.001, and **** *p* < 0.0001 denote statistical significance.

**Figure 6 antioxidants-14-00860-f006:**
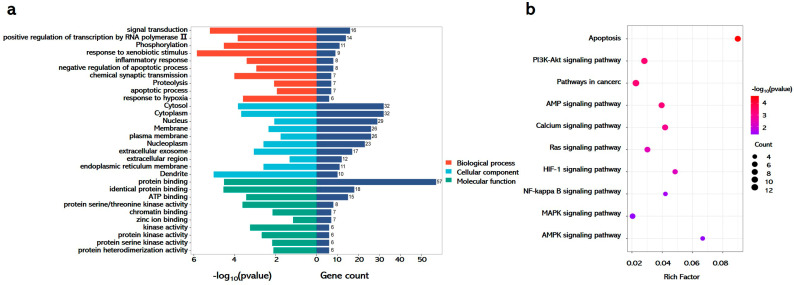
GA against APAP-induced hepatotoxicity through multi-modal mechanisms. (**a**) Go and (**b**) KEGG enrichment analyses were performed using the DAVID database, leveraging shared drug–disease targets as the analytical basis.

**Figure 7 antioxidants-14-00860-f007:**
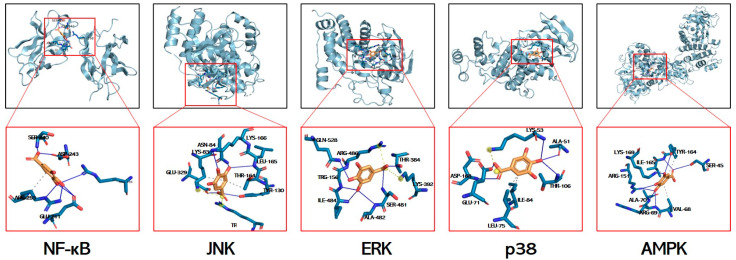
Structural basis of GA’s multi-target engagement against APAP-induced hepatotoxicity. Molecular docking simulation of GA with the NF-κB p65 subunit, JNK, ERK, p38, and AMPK.

**Figure 8 antioxidants-14-00860-f008:**
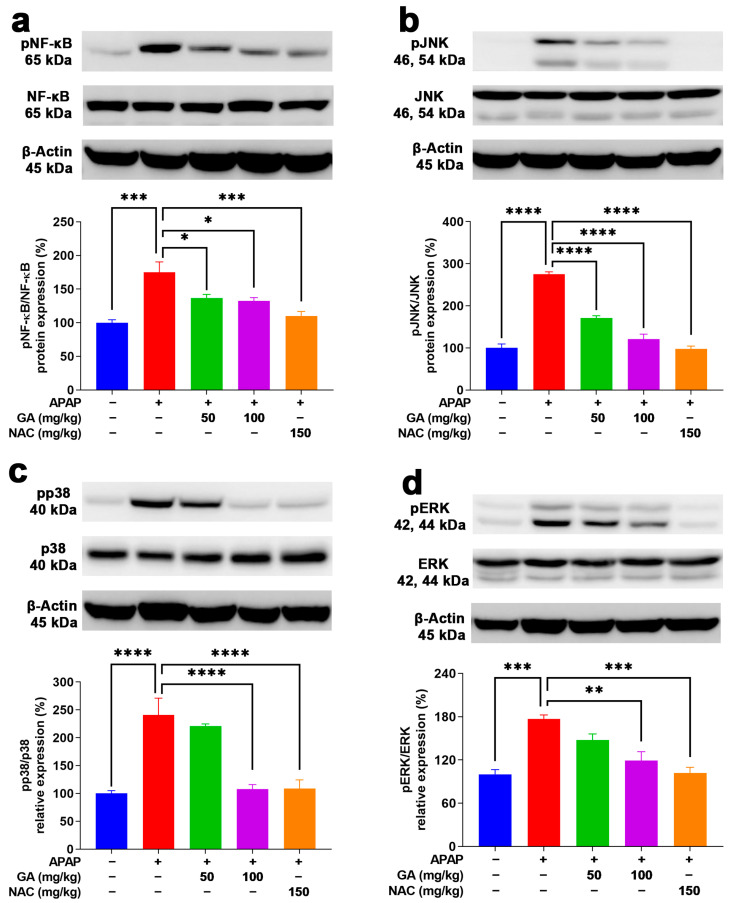
GA treatment reduced the phosphorylation of NF-κB and MAPKs in APAP-induced hepatic inflammation. (**a**) Protein expressions of NF-κB p65, (**b**) JNK, (**c**) p38, and (**d**) ERK AMPK were evaluated. The data are presented as the mean ± standard error of the mean (SEM) (n = 8). * *p* < 0.05, ** *p* < 0.01, *** *p* < 0.001, and **** *p* < 0.0001 denote statistical significance.

**Figure 9 antioxidants-14-00860-f009:**
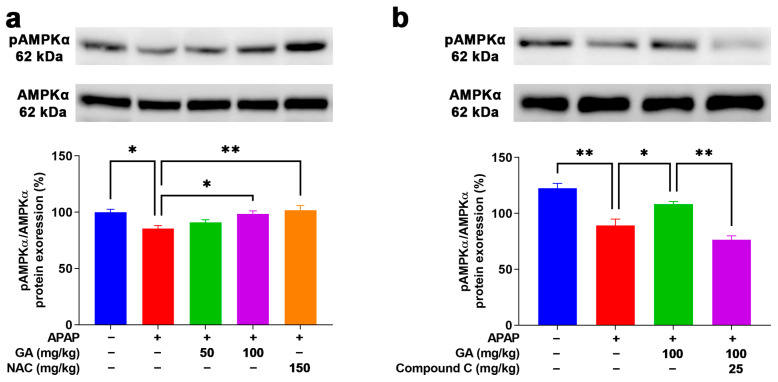
GA attenuated APAP-induced hepatic oxidative stress through AMPK signaling pathways. (**a**,**b**) Immunoblotting was employed to comprehensively detect the expression levels of AMPKα1 proteins both prior to and following the application of AMPK inhibitors. All data are presented as the mean ± SEM (n = 8). * *p* < 0.05 and ** *p* < 0.01 denote statistical significance.

**Table 1 antioxidants-14-00860-t001:** Nucleotide sequence of the primers for qPCR.

Gene Target	Primer		Size	Accession No.
CYP2E1	Forward	GCAGTATCCCACGCGAATTT	184	NM_021282
	Reverse	CCATGTGGTTGCTGGGATTT		
Catalase	Forward	AGGTGTTGAACGAGGAGGAG	196	NM_009804
	Reverse	TGCGTGTAGGTGTGAATTGC		
SOD-1	Forward	ATGGGTTCCACGTCCATCAG	132	NM_011434
	Reverse	CATTGCCCAGGTCTCCAACA		
SOD-2	Forward	GAGAATCTCAGTGCTCACTC	160	NM_013671
	Reverse	GGAACCCTAAATGCTGCCAG		
Bcl-2	Forward	CGGGAGAACAGGGTATGA	149	NM_009741
	Reverse	CAGGCTGGAAGGAGAAGAT		
Bax	Forward	GCAGGGAGGATGGCTGGGGAGA	352	NM_001411995
	Reverse	TCGAGACAAGCAGCCGCTCACG		
GAPDH	Forward	CACTGAGCATCTCCCTCACA	111	NM_008084
	Reverse	GTGGGTGCAGCGAACTTTAT		

**Table 2 antioxidants-14-00860-t002:** Mouse body weight and liver weight were measured, and the liver-to-body weight ratio was calculated. The data are presented as the mean ± standard error of the mean (SEM) (n = 8). * *p* < 0.05, *** *p* < 0.001, and **** *p* < 0.0001 denote statistical significance.

Parameter	Normal Control	APAP	APAP + GA 50	APAP + GA 100	APAP + NAC 150
Body weight (g)	26.15 ± 0.366	26.28 ± 0.237	26.24 ± 0.196	26.22 ± 0.403	26.02 ± 0.430
Liver weight (g)	1.15 ± 0.016	1.29 ± 0.019 ***	1.24 ± 0.014	1.214 ± 0.022 *	1.14 ± 0.022 *
Liver/Body × 100	4.41 ± 0.025	4.92 ± 0.047 ****	4.72 ± 0.041	4.62 ± 0.031 ***	4.60 ± 0.043 ****

**Table 3 antioxidants-14-00860-t003:** Calculated binding energies (kcal/mol) for the molecular docking interactions of GA with NF-κB, JNK, ERK, p38, and AMPK proteins.

Receptor	NF-κB	JNK	ERK	p38	AMPK
Binding energy (kcal/mol)	−3.28	−5.4	−5.6	−6.3	−5.8

## Data Availability

All the data are available within the article.

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
