# Peer review of "Gallic Acid Alleviates Acetaminophen-Induced Acute Liver Injury by Regulating Inflammatory and Oxidative Stress Signaling Proteins"

_antioxidants, 2025, doi:10.3390/antiox14070860_

Round 1

Reviewer 1 Report

Dear Authors,
I have carefully reviewed your manuscript entitled “Gallic acid alleviates acetaminophen-induced acute liver injury by regulating inflammatory and oxidative stress signaling pathways.” Your work presents a timely and scientifically relevant investigation into the hepatoprotective effects of Gallic acid (GA), highlighting its mechanistic involvement in modulating MAPK, NF-κB, and AMPK signaling pathways in a murine model of APAP-induced liver injury.
The experimental design is robust and well-structured, incorporating histological, molecular, and bioinformatic approaches—including KEGG/GO enrichment and molecular docking—which enrich the study’s mechanistic depth. Moreover, the evidence of dose-dependent protective effects of GA is compelling and contributes meaningfully to the field of natural compound-based hepatoprotection.
However, in its current form, the manuscript would benefit from significant linguistic and structural refinement. Several issues related to grammar, syntax, and scientific expression—such as overly literal phrasing, tense inconsistencies, and redundant formulations—affect the clarity and fluency of the narrative. The introduction, in particular, should be restructured to provide a more cohesive and hypothesis-driven rationale for the study. Additionally, the abstract requires revision to improve precision and readability.
Furthermore, mechanistic explanations—particularly concerning the roles of MAPK/AMPK pathways and the use of Compound C—should be articulated more clearly to enhance the scientific impact of your findings. Figure legends and table captions should also be revised for consistency and conciseness, and placeholder elements such as the caption for Table 2 must be addressed.
With careful revision focusing on scientific clarity, linguistic precision, and narrative coherence, your manuscript has strong potential to make a valuable contribution to the literature.
I commend your efforts and encourage you to revise the manuscript accordingly.
Kind regards.

During the review, I highlighted the following critical points

1. Abstract
The abstract is hindered by clumsy syntax and ambiguous phrasing, such as “APAP overdose is a significant driver of drug-induced liver injury (DILI) affecting humans and animals globally, necessitates…,” which should be revised to “which necessitates…,” and “potent medicine properties,” which would be more accurately expressed as “potent medicinal properties.” To enhance clarity and readability, the abstract should be thoroughly revised to streamline its structure, eliminate linguistic imprecision, and improve overall narrative flow.

2. Introduction
The manuscript exhibits overly literal phrasing and awkward transitions that hinder fluency and readability. For instance, expressions such as “APAP administration disrupts this metabolic equilibrium” would benefit from more refined alternatives like “Supratherapeutic doses of APAP disturb hepatic metabolic homeostasis.” Additionally, transitional phrases—such as “While previous studies have explored…”—are often repetitive and lack structural clarity. Paragraphs should flow more cohesively, with smoother logical progression between ideas to enhance narrative continuity.

3. Grammar and Syntax Errors
The manuscript contains numerous grammatical inconsistencies, including frequent comma splices, omission of essential articles (“the,” “a,” “an”), and irregular verb tense usage. These issues are exemplified by phrases such as “GA has demonstrated potent medicine properties,” which should read “potent medicinal properties,” and “Mice were intraperitoneally injected with APAP (300 mg/kg) 1 h before GA,” which would be more fluently expressed as “received an intraperitoneal injection of APAP (300 mg/kg), followed one hour later by GA.” Similarly, constructions like “was used to reverse transcribed” should be corrected to “was used to reverse transcribe.” Addressing these errors would significantly enhance the manuscript’s linguistic precision and overall readability.

4. Scientific Precision and Redundancy
The manuscript would benefit from improved scientific precision and the elimination of redundant phrasing. For example, expressions such as “GA reversed this downward trend of these antioxidant enzymes” are unnecessarily verbose and could be more effectively conveyed as “GA restored the expression of key antioxidant enzymes suppressed by APAP.” Likewise, phrases like “Results can be corroborated through liver weight examination” should be refined to “These findings were corroborated by liver weight measurements,” ensuring greater clarity and conciseness in the scientific narrative.

5. Figures and Tables
The presentation of figures and tables would benefit from greater editorial consistency and conciseness. Many figure captions contain repetitive phrasing, such as “The data are presented as the mean ± SEM...,” which appears unnecessarily across multiple legends and should be streamlined. Additionally, all terminology—such as cytokine names and gene symbols—should be uniformly formatted for clarity and professionalism. Notably, Table 2 currently features a placeholder caption (“This is a table caption”), which should be replaced with a precise and descriptive title to accurately reflect its content.

Conceptual Clarity
The MAPK and AMPK pathways are central to the study's findings, yet their mechanistic roles are insufficiently articulated in the discussion, where pathway names are often repeated without substantive explanation. To enhance clarity, the narrative should explicitly state that GA inhibits MAPK phosphorylation (ERK, JNK, p38), thereby suppressing pro-inflammatory transcription, while concurrently activating AMPK to promote antioxidant gene expression and restore redox homeostasis. Additionally, the role of Compound C as an AMPK inhibitor is underexplained and should be introduced with greater clarity earlier in the Results section to underscore its relevance in validating the pathway's involvement.

Strengths
The study is strengthened by a comprehensive experimental design encompassing histological assessment, gene and protein expression analysis, and detailed pathway investigation. The integration of molecular docking simulations alongside KEGG and GO enrichment analyses adds further depth to the mechanistic insights. Moreover, the data provide clear, dose-dependent evidence of Gallic Acid’s protective effects, reinforcing its therapeutic potential in the context of APAP-induced liver injury.

Final Recommendation
I recommend a major language revision.

1. Abstract (Lines 12–30)
Necessary: Line 12 — The rationale for intervention is underdeveloped. The abstract should better contextualize the therapeutic gap that GA aims to address in relation to NAC's limitations.
Desirable: Line 15 — Replace vague terminology ("potent medicine properties") with specific bioactivities (e.g., antioxidant, anti-inflammatory).

2. Introduction (Lines 33–68)
Necessary: Line 40 — The incidence of APAP-induced ALI is presented generically; epidemiological data or referenced prevalence estimates would strengthen the claim.
Desirable: Line 57 — Clarify AMPK’s role in oxidative stress rather than listing broad cellular functions.

3. Methods
Animal Grouping (Lines 70–84)
Necessary: Line 81 — There is an inconsistency between the abstract and methods regarding the timing of GA/NAC administration ("post" vs. "prior" to APAP). This should be corrected for internal consistency.
Desirable: Justify the selection of the 12-hour endpoint post-APAP exposure.

qPCR (Lines 110–120)

Necessary: Add details regarding primer validation (efficiency, melt curve analysis), and indicate whether technical replicates were performed.

Molecular Docking (Lines 130–139)
Necessary: Clarify how docking accuracy was validated and whether RMSD or similar metrics were used.

4. Results
Liver Weight and Histology (Lines 155–165)

Necessary: Report effect sizes in addition to p-values to enhance interpretability.

Hepatocyte Necrosis (Lines 191–203, Figure 3)
Desirable: Quantify TUNEL-positive areas more explicitly and specify the method of image analysis (manual vs. software-assisted).

Oxidative Stress (Lines 212–231, Figure 4)

Necessary: CYP2E1 mRNA levels are shown, but no protein validation is provided. This is a mechanistically central enzyme and its protein-level expression should be discussed or measured.

Pathway Enrichment (Lines 256–272, Figure 6)

Desirable: The KEGG and GO analyses are presented but not fully interpreted. Highlight the most significantly enriched pathways and relate them back to the biological context.

Molecular Docking (Lines 273–284, Table 2)

Necessary: Table 2 caption currently reads “This is a table caption.” This must be replaced with a meaningful description of the table's content.

5. Discussion (Lines 322–382)
Necessary: The mechanistic explanation of how GA modulates MAPK/AMPK pathways is repetitive and lacks depth. Expand this section with a more integrated interpretation of the signaling cascade.
Necessary: The absence of protein-level CYP2E1 and the lack of pharmacokinetic data for GA should be explicitly acknowledged as limitations.
Desirable: Discuss the translational relevance of GA compared to NAC, especially in terms of therapeutic window and safety.

6. Figures and Tables
Necessary: Table 2 — Placeholder text must be replaced.
Desirable: Figures 4, 5, and 8 should indicate molecular weights and include normalization details for Western blots (e.g., GAPDH).

7. Missing Limitations Section
Necessary: The manuscript lacks a discussion of study limitations. Points to include:

- Use of a single time point (12 h)
- Lack of GA pharmacokinetic and bioavailability data
- Absence of CYP2E1 protein expression data
- Single-sex (male) animal model

Author Response

Dear Reviewer

We sincerely appreciate your time and effort in reviewing our manuscript (ID: antioxidants-3666091), Your insightful comments and constructive suggestions have greatly helped us improve the quality of our work.

We have carefully addressed all of your concerns in the revised manuscript. Detailed point-by-point responses to your comments are provided below. Please let us know if any further clarifications or modifications are needed.

Once again, thank you for your through review and valuable input. Your expertise has been instrumental in shaping this manuscript.

Best regards

Weishun Tian, DVM, PhD
College of Animal Science and Technology
Henan University of Science and Technology
Luoyang 471000, China
Tel: +86-0379-64282341

  1. Abstract
    The abstract is hindered by clumsy syntax and ambiguous phrasing, such as “APAP overdose is a significant driver of drug-induced liver injury (DILI) affecting humans and animals globally, necessitates…,” which should be revised to “which necessitates…,” and “potent medicine properties,” which would be more accurately expressed as “potent medicinal properties.” To enhance clarity and readability, the abstract should be thoroughly revised to streamline its structure, eliminate linguistic imprecision, and improve overall narrative flow.

Response 1: We thank the reviewer for noting the syntax and phrasing issues in our abstract. We have comprehensively revised the section:

(1) Correct the grammatical structure in the highlighted sentence, replacing "necessitates..." with "which necessitates..." (Line 13).

(2) Replace "potent medicine properties" with the accurate term " antioxidant, anti-inflammatory properties " (Line 14-15).

(3) We have restructured sentences to eliminate ambiguity, enhanced narrative flow through logical sequencing of key findings, and resolved all identified linguistic imprecisions (Line 12-28).

  1. Introduction
    The manuscript exhibits overly literal phrasing and awkward transitions that hinder fluency and readability. For instance, expressions such as “APAP administration disrupts this metabolic equilibrium” would benefit from more refined alternatives like “Supratherapeutic doses of APAP disturb hepatic metabolic homeostasis.” Additionally, transitional phrases—such as “While previous studies have explored…”—are often repetitive and lack structural clarity. Paragraphs should flow more cohesively, with smoother logical progression between ideas to enhance narrative continuity.

Response 2: We appreciate the reviewer's astute critique regarding linguistic precision and narrative flow. The Introduction has been comprehensively restructured to:

(1) Replace literal phrasing (e.g., "APAP administration disrupts this metabolic equilibrium") with precise scientific terminology ("Supratherapeutic APAP doses disturb hepatic metabolic homeostasis") (Line 36-37).

(2) Rewritten the sentences to eliminate redundant transitional phrases, enhance inter-sentence coherence (Line 32-69).

  1. Grammar and Syntax Errors
    The manuscript contains numerous grammatical inconsistencies, including frequent comma splices, omission of essential articles (“the,” “a,” “an”), and irregular verb tense usage. These issues are exemplified by phrases such as “GA has demonstrated potent medicine properties,” which should read “potent medicinal properties,” and “Mice were intraperitoneally injected with APAP (300 mg/kg) 1 h before GA,” which would be more fluently expressed as “received an intraperitoneal injection of APAP (300 mg/kg), followed one hour later by GA.” Similarly, constructions like “was used to reverse transcribed” should be corrected to “was used to reverse transcribe.” Addressing these errors would significantly enhance the manuscript’s linguistic precision and overall readability.

Response 3: We thank the reviewer for identifying grammatical inconsistencies. The text has been:

(1) Restructured to eliminate comma splices and standardize articles/verb tenses,

(2) Revised per specific examples (e.g., "medicinal properties", "received intraperitoneal injection").

  1. Scientific Precision and Redundancy
    The manuscript would benefit from improved scientific precision and the elimination of redundant phrasing. For example, expressions such as “GA reversed this downward trend of these antioxidant enzymes” are unnecessarily verbose and could be more effectively conveyed as “GA restored the expression of key antioxidant enzymes suppressed by APAP.” Likewise, phrases like “Results can be corroborated through liver weight examination” should be refined to “These findings were corroborated by liver weight measurements,” ensuring greater clarity and conciseness in the scientific narrative.

Response 4: We thank the reviewer for enhancing our scientific rigor. The manuscript has been comprehensively revised.

(1) Replaced ambiguous expressions with more accurate description.

(2) Optimize the language in the manuscript and mark it with different color. “GA reversed this downward trend of these antioxidant enzymes” to “GA restored the expression of key antioxidant enzymes suppressed by APAP”. “Results can be corroborated through liver weight examination” to “these findings were corroborated by liver weight measurements.”

  1. Figures and Tables
    The presentation of figures and tables would benefit from greater editorial consistency and conciseness. Many figure captions contain repetitive phrasing, such as “The data are presented as the mean ± SEM...,” which appears unnecessarily across multiple legends and should be streamlined. Additionally, all terminology—such as cytokine names and gene symbols—should be uniformly formatted for clarity and professionalism. Notably, Table 2 currently features a placeholder caption (“This is a table caption”), which should be replaced with a precise and descriptive title to accurately reflect its content.

Response 5: We thank the reviewer for enhancing our data presentation rigor. The following comprehensive revisions have been implemented:

(1) unified statistical notation in Methods section.

(2) Standardized terminology per Cell Press style.

(3) replaced Table 2 caption with precise descriptor.

  1. Abstract (Lines 12–30)
    Necessary: Line 12 — The rationale for intervention is underdeveloped. The abstract should better contextualize the therapeutic gap that GA aims to address in relation to NAC's limitations.
    Desirable: Line 15 — Replace vague terminology ("potent medicine properties") with specific bioactivities (e.g., antioxidant, anti-inflammatory).

Response 6: We thank the reviewer for strengthening our abstract's scientific impact. We have given NAC's clinical limitations (Line 15), and added the medicine properties of Gallic acid, such as antioxidant and anti-inflammatory (Line 14).

  1. Introduction (Lines 33–68)
    Necessary: Line 40 — The incidence of APAP-induced ALI is presented generically; epidemiological data or referenced prevalence estimates would strengthen the claim.
    Desirable: Line 57 — Clarify AMPK’s role in oxidative stress rather than listing broad cellular functions.

Response 7: We appreciate the reviewer's insightful suggestions for enhancing scientific rigor. The following targeted revisions have been implemented: (1) Added epidemiological data and the references (Line 38-40), (2) Clarified the AMPK’s role in oxidative stress (Line 57-59).

  1. Methods
    Animal Grouping (Lines 70–84)
    Necessary: Line 81 — There is an inconsistency between the abstract and methods regarding the timing of GA/NAC administration ("post" vs. "prior" to APAP). This should be corrected for internal consistency.
    Desirable: Justify the selection of the 12-hour endpoint post-APAP exposure.

Response 7: We sincerely appreciate the reviewer's meticulous attention to methodological consistency. We have agreed on the timing of the medication in the abstract and methods. Mice received an intraperitoneal injection of APAP (300 mg/kg), followed by oral administration of GA (50 or 100 mg/kg) or NAC (150 mg/kg) 1-hour post-intoxication (Line 17-19 and 80-82). As well we added the reference to justify the selection of the 12-hour endpoint post-APAP exposure (Line 86-87).

qPCR (Lines 110–120)

Necessary: Add details regarding primer validation (efficiency, melt curve analysis), and indicate whether technical replicates were performed.

Response 8: We thank the reviewer for enhancing methodological rigor. The validations have been added to Methods (Line 128-132)

Molecular Docking (Lines 130–139)
Necessary: Clarify how docking accuracy was validated and whether RMSD or similar metrics were used.

Response 9: We sincerely appreciate the reviewer for providing this valuable technical insight. Molecular docking was utilized as a hypothesis-generating approach, and the key findings derived from this method were subsequently validated through in vivo experiments. Additionally, we have incorporated the detailed docking parameters into the Methods section (Lines 148-157). It should be noted that we did not employ root-mean-square deviation (RMSD) validation in this study.

  1. Results
    Liver Weight and Histology (Lines 155–165)

Necessary: Report effect sizes in addition to p-values to enhance interpretability.

Response 10: We thank the reviewer for enhancing our statistical rigor. We have added the weight of the liver in Table 2 of the article to better explain the effect of GA on APAP-induced liver injury. (Line 192-195).

Hepatocyte Necrosis (Lines 191–203, Figure 3)
Desirable: Quantify TUNEL-positive areas more explicitly and specify the method of image analysis (manual vs. software-assisted).

Response 10: We thank the reviewer for this valuable suggestion. The TUNEL quantification methodology has been introduced in Methods (Lines 106-107).

Oxidative Stress (Lines 212–231, Figure 4)

Necessary: CYP2E1 mRNA levels are shown, but no protein validation is provided. This is a mechanistically central enzyme and its protein-level expression should be discussed or measured.

Response 10: Thank s for your comment. Through the protein expression was not measured, the observed upregulation of CYP2E1 mRNA (Fig. 4c) aligns with Elevated serum ALT/AST levels (Fig. 2b), Increased hepatic malondialdehyde (MDA) content (Fig. 4a) and histological evidence of centrilobular necrosis. These endpoints collectively reflect CYP2E1-mediated oxidative stress, as documented in https://pubmed.ncbi.nlm.nih.gov/28668488/.

Moreover, Prior studies have established that APAP hepatotoxicity is predominantly driven by CYP2E1 transcriptionally regulated NAPQI production, justifying our initial approach. https://pmc.ncbi.nlm.nih.gov/articles/PMC5614957/

Pathway Enrichment (Lines 256–272, Figure 6)

Desirable: The KEGG and GO analyses are presented but not fully interpreted. Highlight the most significantly enriched pathways and relate them back to the biological context.

Response 11: We appreciate the reviewer's guidance on enhancing pathway interpretation. KEGG and GO analyses serve as preliminary predictions of the potential mechanisms through which GA alleviates liver damage, yet they cannot substitute for animal experiments. From the perspective of animal welfare, this approach enhances the experimental success rate while significantly reducing the utilization of laboratory animals. Furthermore, GO analysis results indicate that the mechanism of GA in alleviating liver damage is associated with inflammation and apoptosis signaling pathways (Line 284-286), whereas KEGG analysis suggests involvement of the NF-κB/MAPK/AMPK pathways (289-292).

Molecular Docking (Lines 273–284, Table 2)

Necessary: Table 2 caption currently reads “This is a table caption.” This must be replaced with a meaningful description of the table's content.

Response 12: Thank you very much. We are so sorry for this mistake and already replaced with a meaningful description of the table's content (308-309).

  1. Discussion (Lines 322–382)
    Necessary: The mechanistic explanation of how GA modulates MAPK/AMPK pathways is repetitive and lacks depth. Expand this section with a more integrated interpretation of the signaling cascade.
    Necessary: The absence of protein-level CYP2E1 and the lack of pharmacokinetic data for GA should be explicitly acknowledged as limitations.

Desirable: Discuss the translational relevance of GA compared to NAC, especially in terms of therapeutic window and safety.

Response 12: We sincerely appreciate the reviewer's insightful critique.

(1) The Discussion section has been restructured to present an integrated model of GA's coordinated modulation of the AMPK/MAPK/NF-κB pathways, delineating their crosstalk in hepatoprotection (Lines 406-415).

(2) We have explained in “Response 10” and illustrate the current study's limitations including multi-timepoint analyses, pharmacokinetic profiling via LC-MS/MS, CYP2E1 protein quantification, and sex-stratified models in Discuss parts, and further research will be initiated to enhance translational relevance (Line 417-420).

  1. Figures and Tables
    Necessary: Table 2 — Placeholder text must be replaced.
    Desirable: Figures 4, 5, and 8 should indicate molecular weights and include normalization details for Western blots (e.g., GAPDH).

Response 13: We thank the reviewer for identifying these presentation issues. We are so sorry for this mistake and already replaced with a meaningful description of the table's content (306-307). Moreover, we also indicate molecular weights, include normalization details for Western blots (Figure 8).

  1. Missing Limitations Section
    Necessary: The manuscript lacks a discussion of study limitations. Points to include:

- Use of a single time point (12 h)
- Lack of GA pharmacokinetic and bioavailability data
- Absence of CYP2E1 protein expression data
- Single-sex (male) animal model

Response 14: We are truly indebted to you for your invaluable guidance. The APAP-induced liver injury experimental model employed in our study is firmly grounded in prior research endeavors. Specifically, it entails administering APAP to male mice to meticulously investigate the drug's impact on liver injury, a methodology that has been proven reliable in elucidating the underlying mechanisms of such injury. In response to your astute observation, we have incorporated the relevant references in the materials and methods section, ensuring that the experimental setup is thoroughly supported by the existing literature (Line 84-87).

Moreover, we have conducted a comprehensive analysis of the study's limitations, including those related to the model, in the discussion section of the revised manuscript, thereby providing a more complete and nuanced understanding of our research.

Reviewer 2 Report

This paper is potentially interesting but there are some issues that should be carefully addressed by authors before making the paper suitable for publication in the Antioxidants.

Specific comments and suggestions are given below.

Lines 63-64: Please add references regarding previous studies.

Please rewrite Materials and Methods to decrease similarity with https://doi.org/10.1016/j.biopha.2023.115898

Lines 79-84: Please explains dosage for APAP, GA, and NAC.

Line 82: For evaluation of dose-response, at least three doses are needed. Plase explain why did you use only two doses of GA.

Line 86: Please use g, not rpm.

Lines 86-90: Please, briefly describe principles of measurements. The same for all other methods. Quantities of samples are missing.

Figures captions: All abbreviations should be explained (GA, APAP…).

Figures 2a and 3a: Please describe the changes.

Table 2: Table caption is missing.

Lines 327-330: Please add references.

In Discussion section it might be useful to compare/discuss the results with other authors who tested similar natural bioactive compounds. Please make better distinction between your results and results from other studies.

Author Response

Dear Reviewer

We sincerely appreciate your time and effort in reviewing our manuscript (ID: antioxidants-3666091), Your insightful comments and constructive suggestions have greatly helped us improve the quality of our work.

We have carefully addressed all of your concerns in the revised manuscript. Detailed point-by-point responses to your comments are provided below. Please let us know if any further clarifications or modifications are needed.

Once again, thank you for your through review and valuable input. Your expertise has been instrumental in shaping this manuscript.

Best regards

Weishun Tian, DVM, PhD
College of Animal Science and Technology
Henan University of Science and Technology
Luoyang 471000, China
Tel: +86-0379-64282341

Lines 63-64: Please add references regarding previous studies.

Response 1Thank you for pointing this out. We agree with this comment. We have rewritten the sentence (Line 63-64).

Please rewrite Materials and Methods to decrease similarity with https://doi.org/10.1016/j.biopha.2023.115898

Response 2Thank you for pointing this out. We agree with this comment. We have rewritten the Materials and Methods parts to decrease similarity with my previous article.

Lines 79-84: Please explains dosage for APAP, GA, and NAC.

Response 3:We thank the reviewer for this methodological inquiry. The dosages were selected based on established protocols from peer-reviewed studies, we have added the references (Line 82-83)

Line 82: For evaluation of dose-response, at least three doses are needed. Please explain why did you use only two doses of GA.

Response 4We appreciate this insightful methodological query. The selection of two GA doses (50 and 100 mg/kg) was based on the prior pharmacodynamic evidence. https://pubmed.ncbi.nlm.nih.gov/20609067/

Line 86: Please use g, not rpm.

Response 5: Thank your very much! We have corrected “rpm” to “g” (Line 90).

Lines 86-90: Please, briefly describe principles of measurements. The same for all other methods. Quantities of samples are missing.

Figures captions: All abbreviations should be explained (GA, APAP…).

Figures 2a and 3a: Please describe the changes.

Table 2: Table caption is missing.

Lines 327-330: Please add references.

Response 5: We sincerely appreciate the reviewer's meticulous feedback. All concerns have been addressed:

(1) We already describe principles of measurements and added the quantities of samples (Line 94-97, 112-115, 119-121, 136-137).

(2) The abbreviations used in this manuscript are listed at the end of the manuscript (Line 442).

(3) We have described the histopathological changes in Figures 2a and 3a (Line 196-201, 219-223).

(4) We have added the table caption (Line 306-307).

(5) We have added the references (Line 349-352).

Reviewer 3 Report

The objective of the present manuscript was to investigate the molecular mechanisms of the therapeutic potential of gallic acid in Acetaminophen-induced hepatotoxicity.

Although the authors performed different evaluations to elucidate the mechanism of action of gallic acid, it should delve deeper into the methodology used to prepare the treatments, as well as the route of administration.

In addition, the statistical analysis described in the "statistical analysis" section does not agree with the results presented in Figure 1, 2, 3, 4 and 5, since the Tukey test makes a multiple comparison between the groups and the one shown in the figures suggests tests such as Dunnette or Bonferroni given that all are compared only with the positive control

Abstract

L.12. Include the meaning of APAP.
L.14-15. strategies. Gallic acid (GA), a naturally occurring polyphenolic compound widely distributed in plant species, has demonstrated potent medicine properties.
L.16. investigates or investigated????

Introduction

L.67. remove the comma.

Materials and methods

L.70. How were GA and NAC administered? Please detail the methodology. Include GA and NAC specifications. Were solutions administered? How were they dissolved? etc.
L.81-82. 1 h before treatment with GA (50, 100 mg/kg), N-acetylcysteine ​​(NAC) (150 mg/kg).
L.82-83. Mice were anesthetized and sacrificed 12 h after APAP injection.
L.86-90. Please restructure the ideas in this paragraph so that the methodology is better understood.
L.102. which was used.
L.106. Please be more detailed in the methodologies using kits. It would be good to include dilution and detection factors (wavelength).
L.112. Generally, a sentence never begins with a number, and if it does, it should be changed to letters.

Results

L.163-165. Image 1b does not indicate that the protective effects of 100 mg/kg of GA are the same as those of the NAC-treated group. Please include literals or the complete analysis. A Tukey test compares all groups, including the controls. It appears that the authors performed a Dunnette or Bonferroni test, since only the groups are compared with the positive control. Please check.
L.170. Include n=?????
L.180. Serum was collected 12 h after mice were challenged with APAP.
L.187-190. Same comment as in Figure 1. Please check the type of statistical test since it is mentioned that it should be a Tukey test and it is not. Include the number of samples analyzed (n=x).
L.194. After 12 hours of APAP challenge.
L.198. review the writing.
L.213-215. review the writing.
L.216. review the writing.
L.229. when mice were treated only with APAP...
L.230-231. Review this sentence as it is similar to the previous one.

Discussion

L.323. Include the meaning of DILI
L.323. cannot...
L.366. inflammation.
L.369. inflammatory response, and cell death.

Conclusion

L.386. Delete "Mechanistically"

Author Response

Dear Reviewer

We sincerely appreciate your time and effort in reviewing our manuscript (ID: antioxidants-3666091), Your insightful comments and constructive suggestions have greatly helped us improve the quality of our work.

We have carefully addressed all of your concerns in the revised manuscript. Detailed point-by-point responses to your comments are provided below. Please let us know if any further clarifications or modifications are needed.

Once again, thank you for your through review and valuable input. Your expertise has been instrumental in shaping this manuscript.

Best regards

Weishun Tian, DVM, PhD
College of Animal Science and Technology
Henan University of Science and Technology
Luoyang 471000, China
Tel: +86-0379-64282341

Abstract

L.12. Include the meaning of APAP.
L.14-15. strategies. Gallic acid (GA), a naturally occurring polyphenolic compound widely distributed in plant species, has demonstrated potent medicine properties.
L.16. investigates or investigated????

Response 1: Thank you for pointing this out. We agree with this comment. We have added the meaning of APAP (Line 12), and the medicine properties of Gallic acid (GA) that has demonstrated (Line 14-15), and corrected the Grammar error (Line 16).

Introduction

L.67. remove the comma.

Response 2: Thank you for pointing this out. We agree with this comment. We have removed the comma (Line 67).

Materials and methods

L.70. How were GA and NAC administered? Please detail the methodology. Include GA and NAC specifications. Were solutions administered? How were they dissolved? etc.

Response 3: Thank you for raising this important point. We agree that additional methodological detail is warranted. In response, we have explained explicitly detail the administration method for GA and NAC. This now includes the specifications of both compounds, confirmation that they were administered as solutions dissolved in saline, and a description of the dissolution procedure prior to administration (Line 80-83).

L.81-82. 1 h before treatment with GA (50, 100 mg/kg), N-acetylcysteine ​​(NAC) (150 mg/kg).
L.82-83. Mice were anesthetized and sacrificed 12 h after APAP injection.
L.86-90. Please restructure the ideas in this paragraph so that the methodology is better understood.
L.102. which was used.
L.106. Please be more detailed in the methodologies using kits. It would be good to include dilution and detection factors (wavelength).
L.112. Generally, a sentence never begins with a number, and if it does, it should be changed to letters.

Response 4: We thank the reviewer for their valuable feedback. We have carefully addressed each point raised:

L.81-82 & L.82-83 (Timing): We have revised the text to explicitly state the timing sequence: GA or NAC was administered 1 hour after APAP injection, and mice were sacrificed 12 hours after APAP injection (Lines 83-84, 88).

L.86-90: The paragraph describing the experimental timeline, treatments, and sacrifice has been restructured for improved chronological clarity and logical flow (Lines 83-88).

L.102: The phrase flagged as "which was used" has been revised to ensure grammatical correctness and clarity (Line 109).

L.106: Some detailed assay procedures have been added, including specific absorbance wavelengths (Line 94, 113, 119).

L.112: The sentence beginning with a number has been rephrased (Line 127).

Results

L.163-165. Image 1b does not indicate that the protective effects of 100 mg/kg of GA are the same as those of the NAC-treated group. Please include literals or the complete analysis. A Tukey test compares all groups, including the controls. It appears that the authors performed a Dunnette or Bonferroni test, since only the groups are compared with the positive control. Please check.
L.170. Include n=?????
L.180. Serum was collected 12 h after mice were challenged with APAP.
L.187-190. Same comment as in Figure 1. Please check the type of statistical test since it is mentioned that it should be a Tukey test and it is not. Include the number of samples analyzed (n=x).
L.194. After 12 hours of APAP challenge.
L.198. review the writing.
L.213-215. review the writing.
L.216. review the writing.
L.229. when mice were treated only with APAP...
L.230-231. Review this sentence as it is similar to the previous one.

Response 5: We sincerely appreciate the reviewer's meticulous examination of our manuscript. All raised points have been comprehensively addressed in the revised manuscript:
L.163-165, L.187-190: We acknowledge the need for greater transparency in statistical reporting. The Tukey's test was correctly applied for all inter-group comparisons. Non-significant results (*p* ≥ 0.05) were intentionally omitted from figure labeling to enhance visual clarity, as now explicitly stated in the Data Analysis section (Lines 172-173).
L.170, l.187-189: The sample size (*n* = 8 per group) has been added to relevant figure legends (Line 190 and corresponding figure captions).
L.180, L.194, L.198, L.213-215, L.216: We have rewritten the sentences (Line 206-207), (Line 219), (Line 224), (Line 239-241).
L.229, L.230-231. We have corrected the sentence (Line 254-256).

Discussion

L.323. Include the meaning of DILI
L.323. cannot...
L.366. inflammation.
L.369. inflammatory response, and cell death.

Conclusion

L.386. Delete "Mechanistically"

Response 6: We are grateful to the reviewer for their thorough evaluation. The following revisions have been implemented:

L.323. We have added the meaning of DILI (Line 345).
L.366, L.369: We have corrected (Line 394, 401-402, 416).

L.386: We have deleted "Mechanistically" (Line 416).

Round 2

Reviewer 1 Report

Dear Authors,

I have carefully reviewed your manuscript and commend the relevance of the study, which addresses the pressing issue of acetaminophen (APAP) overdose—a leading cause of drug-induced liver injury (DILI). This study provides strong evidence for the hepatoprotective potential of gallic acid (GA) in a murine model, demonstrating reductions in liver damage, enhanced antioxidant defenses, and modulation of MAPK and AMPK pathways—comparable in efficacy to N-acetylcysteine (NAC). These mechanistic findings further underscore GA’s promise as a therapeutic agent.

Nonetheless, the manuscript would be strengthened by a more explicit contextualization within the current body of literature, a more measured presentation of mechanistic interpretations, and a succinct acknowledgment of the study’s limitations. Furthermore, refinement of the language is needed to improve clarity, precision, and scholarly tone.
With these enhancements, the work holds considerable promise to make a valuable contribution to the field. During my review, I identified the following key concerns:

Abstract (Page 1, Lines 12–29)

Line 12: The phrase “necessitates the development” would be more appropriately rendered as “necessitating the development” to ensure grammatical correctness and improve flow.

Line 14: The expression “potent medicine properties” should be revised to “potent medicinal properties” for greater linguistic precision and academic tone.

Lines 26–29: The mention of MAPK and AMPK pathways is introduced rather abruptly, without sufficient contextual grounding. It is recommended to provide a brief rationale or transitional phrasing to better integrate these mechanisms within the abstract’s narrative.

Recommendations:

The abstract would benefit from refined sentence structures to enhance clarity and fluency, a more coherent introduction of GA’s role prior to discussing mechanistic insights, and a restructured concluding sentence to improve logical flow and overall readability.

  1. Introduction (Pages 2–3, Lines 41–68)

Lines 41–47: The detailed exposition of APAP metabolism and oxidative stress mechanisms may be overly technical for readers less familiar with the subject matter.

Lines 48–56: Some mechanistic elements are repeated without offering substantive new insights, which may affect the conciseness of the narrative.

Lines 60–65: The transition from the discussion of APAP toxicity to the introduction of GA’s therapeutic potential would benefit from improved narrative cohesion and flow.

Recommendations:

The introduction would benefit from a streamlined biochemical background, a more cohesive presentation of GA as a potential therapeutic agent, and the use of more varied and precise vocabulary to reduce redundancy and enhance clarity.

  1. Materials and Methods (Page 3, Lines 71–84)

This section would benefit from consistent use of verb tenses, more ethically appropriate phrasing—such as revising “mice were anesthetized and sacrificed” to “euthanized under anesthesia” (Line 83)—and a more comprehensive description of the statistical methods, including details such as confidence intervals and effect sizes.

Recommendations:

To enhance clarity and rigor, this section should maintain consistent use of the past tense, adopt ethically appropriate language when describing animal procedures, and provide more detailed statistical reporting to support reproducibility.

Results (Pages 4–12)

The results section would benefit from more precise and varied language, as terms like “markedly” and “notably” are overused; additionally, some findings lack sufficient contextual explanation, and vague expressions such as “mitigates liver burden” should be clarified by specifying the type of burden—whether histological, biochemical, or otherwise.

Section 3.1–3.2: Elucidate the causal relationship between GA administration and the observed reduction in liver size and tissue damage.

Section 3.3 (Page 5, Lines 21–22): As TUNEL staining specifically detects apoptotic cell death rather than necrosis, the terminology should be revised accordingly to ensure accuracy.

Section 3.4: The repeated use of the phrase “GA reversed the downward trend” could be streamlined for conciseness and stylistic refinement.

Sections 3.5–3.7: The interpretation of molecular docking results would benefit from a more critical discussion of their potential limitations.

Sections 3.8–3.9: The mechanistic conclusions are conveyed with excessive certainty and would benefit from more cautious and appropriately qualified interpretation.

Recommendations:

To enhance clarity and precision, the manuscript should incorporate more varied language to avoid repetition, ensure the accurate use of technical terminology, and explicitly acknowledge the limitations of the proposed mechanistic interpretations.

  1. Discussion (Pages 13–14)

The discussion section would benefit from a more analytical approach, as key findings are frequently restated without added interpretive depth, comparisons with existing literature are limited, and the coverage of molecular pathways is uneven—some are explored in detail while others receive only cursory mention.

Recommendations:

The discussion should offer a more concise summary of the findings, with greater emphasis on their interpretation; it should also include a more critical evaluation of how the results align with or differ from existing literature, while ensuring a balanced treatment of all investigated mechanisms.

  1. Conclusion (Page 15)
  • The repeated use of the term “Mechanistically” within a single sentence should be avoided for stylistic clarity, and the reference to “further translational research” would benefit from greater specificity to enhance its impact.

Recommendations:

The phrasing should be refined for greater clarity and conciseness, and the manuscript should specify the types of translational research warranted—such as pharmacokinetic analyses or in vivo efficacy studies—to provide clearer direction for future work.

Language and Style

The manuscript contains minor grammatical and syntactic errors throughout, along with redundant expressions that detract from stylistic precision; additionally, several figure legends and table captions would benefit from enhanced clarity and editorial refinement.

Recommendation:

A comprehensive language revision by a qualified scientific editor with expertise in academic English would substantially enhance the clarity and overall quality of the manuscript.

The following is a focused evaluation of the manuscript’s scientific content, referencing specific line numbers, tables, and figures where relevant. This assessment clearly distinguishes between essential corrections and recommended improvements, while intentionally excluding comments related to grammar, spelling, or formatting.

  1. Line 21–22 (TUNEL staining interpretation) – Necessary

The manuscript inaccurately associates TUNEL staining with necrosis, whereas this technique is widely recognized for its specificity in detecting apoptotic cell death. This conceptual error should be addressed to ensure scientific accuracy in describing the mode of cell death evaluated.

Correction: Any mention of necrosis in relation to TUNEL staining should be thoroughly reviewed and revised to reflect its well-established specificity for detecting apoptosis, ensuring consistent and accurate usage throughout both the Results and Discussion sections.

  1. Lines 41–47 (Biochemical detail) – Desirable

The early sections of the introduction present a dense mechanistic overview of APAP metabolism and oxidative stress. Although scientifically sound, this level of detail may obscure the central narrative and limit accessibility for a broader scientific readership.

Improvement: It is advisable to streamline this section to enhance clarity and better sustain the reader’s focus on the underlying rationale for exploring gallic acid as a therapeutic candidate.

  1. Lines 48–56 (Repetition of mechanisms) – Desirable

The repeated discussion of oxidative stress and inflammatory pathways, without introducing additional insights, diminishes the scientific conciseness and dilutes the impact of the narrative.

Improvement: Consider restructuring the content to eliminate redundancy and more effectively highlight the unique contributions of gallic acid in modulating these signaling pathways.

  1. Lines 60–65 (Transition to GA) – Necessary

The introduction of gallic acid lacks a smooth transition and is insufficiently framed within the context of established hepatoprotective agents and prior research on polyphenolic compounds.

Correction: To enhance coherence, consider introducing gallic acid with a more seamless transition, supported by a concise overview of relevant literature on its hepatoprotective and pharmacological properties.

  1. Figure Legends – Necessary

A number of figure legends lack clear definitions of symbols denoting statistical significance (e.g., asterisks) and omit essential details such as sample sizes or the number of replicates (n-values), which are critical for interpretability.

Correction: Each figure legend should include:

  • Clear explanation of symbols (p-values)
  • Sample size (n) per group
  • Full definition of any abbreviations used
  1. Molecular Docking (Section 3.5–3.7) – Desirable

While the docking results are presented favorably, the absence of a critical appraisal of inherent limitations—such as binding site flexibility, solvent interactions, and translational relevance to in vivo conditions—detracts from the scientific rigor of the analysis.

Improvement: It is important to recognize that molecular docking serves as a predictive tool rather than a conclusive method, and its inherent limitations should be explicitly acknowledged.

  1. Use of Compound C (AMPK inhibitor) – Necessary

While Compound C is employed to demonstrate AMPK pathway involvement, the manuscript does not address its known off-target effects or the absence of complementary validation techniques, such as Western blot analysis for phospho-AMPK, which are essential for substantiating pathway specificity.

Correction: Consider incorporating a statement in the discussion acknowledging that Compound C may exert off-target effects on other kinases, and that supplementary validation—such as protein-level assays—would enhance the robustness of the proposed mechanistic interpretation.

**8. Statistical Methods (Lines 84 onward) – Potentially Necessary

The statistical analysis section lacks specific details on:

  • Tests used for normality
  • Type of ANOVA (e.g., one-way, two-way)
  • Post hoc test details
  • Clarify whether adjustments for multiple comparisons were applied to control for type I error
  • Recommendation: Given the apparent presence of multiple comparisons in this study, it would be advisable to consult a statistician to ensure the appropriateness of the analytical approach, with particular attention to the following aspects:
  • Correction methods (e.g., Bonferroni, Tukey)
  • Effect sizes and confidence intervals, if relevant
  • Justification for sample size per group (power analysis)
  1. Lines 249–252 (Conclusion) – Necessary

While the assertion that “GA confers protection through modulation of oxidative stress, inflammation, and apoptosis” is generally supported by the data, it conveys a degree of mechanistic certainty that may exceed the evidence presented.

Correction: The mechanistic interpretation should be tempered to reflect that, although the associative data are suggestive, definitive causal relationships remain unconfirmed in the absence of protein-level validation or gene-silencing approaches.

  1. Table 1 (Biochemical Parameters) – Desirable

Table 1 presents key serum biomarkers; however, it omits units in the column headers and displays inconsistencies in numerical precision, such as irregular use of decimal places, which may hinder interpretability and reproducibility.

Improvement: Verify that all reported measurements are accompanied by appropriate units (e.g., U/L) and are presented using uniform decimal formatting to ensure clarity and consistency.

Author Response

Dear Reviewer,

We are profoundly grateful for your continued guidance during the peer review process of our manuscript (ID: antioxidants-3666091). Your expertise and meticulous critiques have been invaluable in elevating the scientific rigor of this work.

We have incorporated all suggestions from your latest review with utmost care:

  • Implemented point-by-point revisions in the manuscript
  • Provided detailed responses to each new comment below

Your dedication to enhancing this research is deeply appreciated. Should any additional refinements be needed, we stand ready to address them promptly.

With sincere respect for your scholarly contribution,

Weishun Tian, DVM, PhD
College of Animal Science and Technology
Henan University of Science and Technology
Luoyang 471000, China
Tel: +86-379-64282341

Detailed comments

  1. Line 21–22 (TUNEL staining interpretation) – Necessary

The manuscript inaccurately associates TUNEL staining with necrosis, whereas this technique is widely recognized for its specificity in detecting apoptotic cell death. This conceptual error should be addressed to ensure scientific accuracy in describing the mode of cell death evaluated.

Correction: Any mention of necrosis in relation to TUNEL staining should be thoroughly reviewed and revised to reflect its well-established specificity for detecting apoptosis, ensuring consistent and accurate usage throughout both the Results and Discussion sections.

Response 1: We sincerely appreciate this critical correction regarding TUNEL methodology. We have revised all erroneous associations between TUNEL and necrosis throughout the manuscript and corrected (Line 22, 223-228)

  1. Lines 41–47 (Biochemical detail) – Desirable

The early sections of the introduction present a dense mechanistic overview of APAP metabolism and oxidative stress. Although scientifically sound, this level of detail may obscure the central narrative and limit accessibility for a broader scientific readership.

Improvement: It is advisable to streamline this section to enhance clarity and better sustain the reader’s focus on the underlying rationale for exploring gallic acid as a therapeutic candidate.

Response 2: We thank the reviewer for the constructive suggestion to enhance narrative clarity. We have greatly simplified the introduction of the APAP metabolic section (Line 41-46) and strengthened the GA research (Line 57-60).

  1. Lines 48–56 (Repetition of mechanisms) – Desirable

The repeated discussion of oxidative stress and inflammatory pathways, without introducing additional insights, diminishes the scientific conciseness and dilutes the impact of the narrative.

Improvement: Consider restructuring the content to eliminate redundancy and more effectively highlight the unique contributions of gallic acid in modulating these signaling pathways.

Response 3: We sincerely thank the reviewer for the comments to this section. We have restructured this section by removing redundant expressions and highlighting the unique mechanism of GA (Line 47-60).

  1. Lines 60–65 (Transition to GA) – Necessary

The introduction of gallic acid lacks a smooth transition and is insufficiently framed within the context of established hepatoprotective agents and prior research on polyphenolic compounds.

Correction: To enhance coherence, consider introducing gallic acid with a more seamless transition, supported by a concise overview of relevant literature on its hepatoprotective and pharmacological properties.

Response 4: We thank the reviewer for this constructive suggestion. We have added the relevant literatures and restructured the sentences to enhance the manuscript coherence (Line 62-67, References 23, 24).

  1. Figure Legends – Necessary

A number of figure legends lack clear definitions of symbols denoting statistical significance (e.g., asterisks) and omit essential details such as sample sizes or the number of replicates (n-values), which are critical for interpretability.

Correction: Each figure legend should include:

  • Clear explanation of symbols (p-values)
  • Sample size (n) per group
  • Full definition of any abbreviations used

Response 5: We sincerely appreciate the reviewer's meticulous feedback on figure annotation deficiencies. We have declared sample sizes per experimental group, and explained the symbols (p-values). (Line 199, 220, 220, 221, 241, 242, 264, 282, 283, 332, 348)

Abbreviations are defined at initial mention in the text, with a master glossary provided in Abbreviations (Line 450). Readers can refer to this resource for all abbreviated terms used throughout the study.

  1. Molecular Docking (Section 3.5–3.7) – Desirable

While the docking results are presented favorably, the absence of a critical appraisal of inherent limitations—such as binding site flexibility, solvent interactions, and translational relevance to in vivo conditions—detracts from the scientific rigor of the analysis.

Improvement: It is important to recognize that molecular docking serves as a predictive tool rather than a conclusive method, and its inherent limitations should be explicitly acknowledged.

Response 6: We sincerely appreciate the reviewer's insightful critique regarding the need to address molecular docking limitations. In response, we have incorporated a dedicated limitations subsection (Lines 302-304) explicitly acknowledging inherent constraints of the computational approach.

  1. Use of Compound C (AMPK inhibitor) – Necessary

While Compound C is employed to demonstrate AMPK pathway involvement, the manuscript does not address its known off-target effects or the absence of complementary validation techniques, such as Western blot analysis for phospho-AMPK, which are essential for substantiating pathway specificity.

Correction: Consider incorporating a statement in the discussion acknowledging that Compound C may exert off-target effects on other kinases, and that supplementary validation—such as protein-level assays—would enhance the robustness of the proposed mechanistic interpretation.

Response 7: We sincerely appreciate the reviewer's insightful critique regarding Compound C's specificity validation. In this study, Compound C was employed to inhibit AMPK activation based on its established mechanism. The observed functional outcomes align with prior reports of AMPK-dependent cytoprotection. We have added the references in Discussion (Line 418, Reference 15,55)

  1. **8. Statistical Methods (Lines 84 onward) – Potentially Necessary

The statistical analysis section lacks specific details on:

  • Tests used for normality
  • Type of ANOVA (e.g., one-way, two-way)
  • Post hoc test details
  • Clarify whether adjustments for multiple comparisons were applied to control for type I error
  • Recommendation: Given the apparent presence of multiple comparisons in this study, it would be advisable to consult a statistician to ensure the appropriateness of the analytical approach, with particular attention to the following aspects:
  • Correction methods(e.g., Bonferroni, Tukey)
  • Effect sizes and confidence intervals, if relevant
  • Justification for sample size per group (power analysis)

Response 8: We sincerely appreciate the reviewer's meticulous statistical critique. Group differences were evaluated by one-way analysis of variance (ANOVA) followed by Tukey’s test according previous studies

  1. Lines 249–252 (Conclusion) – Necessary

While the assertion that “GA confers protection through modulation of oxidative stress, inflammation, and apoptosis” is generally supported by the data, it conveys a degree of mechanistic certainty that may exceed the evidence presented.

Correction: The mechanistic interpretation should be tempered to reflect that, although the associative data are suggestive, definitive causal relationships remain unconfirmed in the absence of protein-level validation or gene-silencing approaches.

Response 9: We sincerely appreciate the reviewer's comment. We have reorganized our expression (Line 254-255).

  1. Table 1 (Biochemical Parameters) – Desirable

Table 1 presents key serum biomarkers; however, it omits units in the column headers and displays inconsistencies in numerical precision, such as irregular use of decimal places, which may hinder interpretability and reproducibility.

Improvement: Verify that all reported measurements are accompanied by appropriate units (e.g., U/L) and are presented using uniform decimal formatting to ensure clarity and consistency.

Response 10: We sincerely express our gratitude for the reviewer's valuable suggestion. In the context of APAP-induced liver injury, the biochemical indicators undergo a substantial elevation, whereas the baseline levels in normal conditions are relatively low, resulting in a pronounced disparity between the two groups. We conducted multiple trials with alternative methods; however, these attempts led to nearly undisplay biochemical indicators in the control group, thereby compromising the interpretability and validity of the results.

Reviewer 2 Report

There are some minor issues that should be addressed by authors.

Line 90: rpm and g cannot be the same numbers. You have to use rotor radius for calculation.

Figures 2a and 3a: Please describe the changes that you marked with black rectangles.

Author Response

Dear Reviewer, We are profoundly grateful for your ongoing expert guidance throughout the revision cycles of our manuscript (ID: antioxidants-3666091). Your incisive critiques have been instrumental in refining the scientific rigor and clarity of this work. Your dedication to scholarly excellence has elevated this research to new standards. Should any final refinements be needed, we remain prepared to address them promptly. With profound appreciation for your commitment to scientific excellence, Weishun Tian, DVM, PhD College of Animal Science and Technology Henan University of Science and Technology Luoyang 471000, China T: +86-379-6428 2341   Line 90: rpm and g cannot be the same numbers. You have to use rotor radius for calculation. Response 1: We deeply appreciate your expert identification of the centrifugation parameter discrepancy. Upon re-examining our experimental records and centrifuge settings, we confirm the correct parameter for serum separation was 3,000 × g (not rpm), as originally programmed in the centrifuge unit. Figures 2a and 3a: Please describe the changes that you marked with black rectangles. Response 2: We thank the reviewer for requesting clarification of the annotated region. We have described the changes in Line 204-208, and Line 225-229.

Reviewer 3 Report

Although the manuscript improved in quality because the authors considered each of the suggestions made in the first revision, the statistical analysis shown in figures or tables does not correspond to a Tukey test. In the Tukey test, all treatments are compared and not just against the control. Please perform the corresponding analysis.

L.339. responses.

Author Response

We are profoundly grateful for your continued guidance during the peer review process of our manuscript (ID: antioxidants-3666091). Your expertise and meticulous critiques have been invaluable in elevating the scientific rigor of this work.

We have incorporated all suggestions from your latest review with utmost care:

  • Implemented point-by-point revisions in the manuscript
  • Provided detailed responses to each new comment below

Your dedication to enhancing this research is deeply appreciated. Should any additional refinements be needed, we stand ready to address them promptly.

With sincere respect for your scholarly contribution,

Weishun Tian, DVM, PhD
College of Animal Science and Technology
Henan University of Science and Technology
Luoyang 471000, China
Tel: +86-379-64282341

Major comments

Although the manuscript improved in quality because the authors considered each of the suggestions made in the first revision, the statistical analysis shown in figures or tables does not correspond to a Tukey test. In the Tukey test, all treatments are compared and not just against the control. Please perform the corresponding analysis.

Response 1: We sincerely appreciate the reviewer's meticulous statistical critique. Our analysis did employ Tukey's test following one-way ANOVA, as evidenced by the software output. To address the specific hypotheses regarding GA and liver protection, this study highlighted the injuries induced by APAP (control group vs. APAP group) as well as the treatment effects (APAP group vs. treatment group), while the comparison results between other groups were not presented. We have made further clarifications in the manuscript.

Detailed comments

L.339. responses.

Response 2: We sincerely apologize for such an error, and have corrected.

Round 3

Reviewer 1 Report

Positive Contributions and Strengths:

  1. This study responds to the urgent demand for novel or complementary therapeutic strategies to N-acetylcysteine (NAC)—the current first-line treatment for acetaminophen (APAP) overdose—whose clinical utility is constrained by a narrow therapeutic window and undesirable side effects.
  2. The manuscript offers a comprehensive mechanistic investigation, incorporating biochemical markers (ALT, AST, MDA, GSH), gene expression profiling of apoptotic and antioxidant regulators (such as Bcl-2, Bax, and key antioxidant enzymes), analysis of protein signaling pathways (including MAPKs, NF-κB, and AMPK), and molecular docking simulations. This integrative approach robustly reinforces the conclusion that gallic acid (GA) exerts its hepatoprotective effects through modulation of oxidative stress and inflammatory responses.
  3. The use of N-acetylcysteine (NAC) as a positive control represents a methodological advantage, enabling a direct and meaningful comparison to assess the therapeutic efficacy of gallic acid (GA).
  4. The GO and KEGG pathway enrichment analyses provide valuable translational and systems-level insights, reinforcing the biological credibility of gallic acid’s molecular targets.

Potential for Improvement and Future Directions:

  1. A key shortcoming of the discussion is the absence of a critical appraisal of the study’s limitations, including the exclusive reliance on a single animal model without human validation, the relatively short observation window of 12 hours, and the lack of pharmacokinetic data or safety profiling for gallic acid at the tested doses.
  2. Although the study presents encouraging preclinical results, it falls short of addressing several critical translational considerations, such as the bioavailability of gallic acid, its potential for drug interactions or adverse effects under long-term or high-dose exposure, and the practical feasibility of its clinical formulation and application.
  3. Several aspects—most notably those concerning inflammation and oxidative stress—are reiterated throughout the results and discussion sections. Streamlining these segments would enhance the narrative's clarity, coherence, and conciseness.
  4. Presentation Quality:
    As the peer-review version does not include the actual figures, their visual quality and clarity cannot be assessed. Additionally, Table 2 lacks a definitive caption. Both elements should be addressed to ensure compliance with the journal’s publication standards.

Relevance in Current Context:

The exploration of naturally occurring bioactive substances with anti-inflammatory and antioxidant effects is an expanding area of research. Gallic acid has already demonstrated its efficacy in various disease contexts. This study broadens its significance by demonstrating its protective role against liver damage caused by acetaminophen (APAP), providing crucial mechanistic evidence. These findings contribute to the growing interest in plant-based complementary treatments for liver disorders.

Necessary Corrections

  1. Lines 273–279 / Figure 7 / Table 2: The reported docking scores (approximately –5 to –6 kcal/mol) indicate a plausible affinity between gallic acid (GA) and molecular targets such as MAPKs and AMPK. However, these values alone are insufficient to confirm functional binding or biological activity. Therefore, the assertion that “GA can bind to JNK, ERK, p38, AMPK, and NF-κB to attenuate APAP-induced acute liver injury” overstates the evidence and should be reframed as a theoretical possibility or presented with greater caution to reflect the preliminary nature of in silico findings.
  2. Line 81 (experimental design): While NAC was appropriately employed as a reference compound, it is important to recognize that its pharmacokinetic properties and mechanism of action—as a glutathione precursor—differ fundamentally from those of gallic acid. Although the comparison provides valuable contextual insight, it does not establish therapeutic equivalence, particularly given the disparities in metabolic pathways and timing of administration. This methodological limitation warrants explicit acknowledgment and discussion in the Discussionsection.
  3. Lines 285–296 / Figure 8: The Western blot results, while visually suggestive, are not accompanied by quantitative analysis such as densitometry, which is essential to substantiate claims of significant modulation of NF-κB and MAPK protein levels. The absence of such quantitative validation diminishes the strength of the mechanistic interpretations presented.
  4. Lines 191–203 / Figure 3: TUNEL staining identifies DNA fragmentation, a hallmark of both apoptotic and necrotic cell death. However, the authors attribute the observed TUNEL-positive signals exclusively to apoptosis, which is not accurate within the context of APAP-induced liver injury, where necrotic mechanisms are also prominent. To prevent potential misinterpretation, this point should be clarified and the dual relevance of TUNEL staining acknowledged.
  5. Lines 285–303 / Figure 8: Although GA treatment appears to diminish the phosphorylation levels of MAPKs and NF-κB, this observation alone does not conclusively demonstrate functional pathway inhibition. Definitive evidence would require analysis of downstream effectors or the use of pathway-specific reporter assays. Accordingly, the current interpretation should be tempered to more accurately reflect the limitations of the data presented.

Desirable Improvements

  1. Lines 309–312 / Figure 9: The involvement of AMPK was explored through the use of Compound C; however, this inhibitor is documented to exert off-target effects that may confound interpretation. It is therefore important that the limitations associated with the specificity of this pharmacological tool be clearly acknowledged and discussed.
  2. Line 83: Tissue collection was performed 12 hours after APAP administration, which is a generally acceptable timepoint in this model. However, it would be beneficial to specify whether this interval corresponds to the peak of hepatic injury and maximal inflammatory response, as such temporal alignment is critical for accurately interpreting the observed pathological and molecular outcomes.
  3. Lines 130–139: The robustness of the molecular docking analysis could be significantly enhanced by incorporating known ligands for each target protein as positive controls. This would provide a valuable reference framework, allowing for a more meaningful interpretation of GA’s binding affinities in relation to established ligand-target interactions.
  4. Lines 81–82: The manuscript does not provide a clear rationale for selecting the 50 and 100 mg/kg doses of gallic acid. It would be important to clarify whether these dosing regimens were informed by prior studies on pharmacokinetics, toxicity, or therapeutic efficacy. Including such justification would enhance the translational significance and scientific credibility of the dosing strategy employed.

Statistical Review Required?

A thorough statistical review is strongly advised to ensure the analytical integrity and validity of the study’s findings. Particular scrutiny should be directed toward Figures 1–9 to confirm the correct implementation of one-way ANOVA and subsequent Tukey’s post-hoc analyses across all experimental groups and outcome measures. Additionally, it is crucial to verify whether foundational statistical assumptions—specifically, data normality and homogeneity of variances—were appropriately assessed and satisfied. Given the considerable number of simultaneous comparisons performed (e.g., cytokine profiles, gene expression data, enzymatic activity), the use of multiple comparison adjustments—such as the Bonferroni correction—should be considered to mitigate the risk of type I statistical errors.

Author Response

Dear Reviewer, We are profoundly grateful for your ongoing expert guidance throughout the revision cycles of our manuscript (ID: antioxidants-3666091). Your incisive critiques have been instrumental in refining the scientific rigor and clarity of this work. Your dedication to scholarly excellence has elevated this research to new standards. Should any final refinements be needed, we remain prepared to address them promptly. With profound appreciation for your commitment to scientific excellence, Weishun Tian, DVM, PhD College of Animal Science and Technology Henan University of Science and Technology Luoyang 471000, China T: +86-379-6428 2341   Necessary Corrections 1. Lines 273–279 / Figure 7 / Table 2: The reported docking scores (approximately –5 to –6 kcal/mol) indicate a plausible affinity between gallic acid (GA) and molecular targets such as MAPKs and AMPK. However, these values alone are insufficient to confirm functional binding or biological activity. Therefore, the assertion that “GA can bind to JNK, ERK, p38, AMPK, and NF-κB to attenuate APAP-induced acute liver injury” overstates the evidence and should be reframed as a theoretical possibility or presented with greater caution to reflect the preliminary nature of in silico findings. Response 1: We sincerely appreciate the reviewer's crucial insight regarding the interpretation of docking results. We have implemented the following revisions “Molecular docking predicts the potential binding of GA to JNK/ERK/p38/AMPK/NF-κB, and is expected to be a mechanism for resolving GA in alleviating liver injury” (Line 310-312) 2. Line 81 (experimental design): While NAC was appropriately employed as a reference compound, it is important to recognize that its pharmacokinetic properties and mechanism of action—as a glutathione precursor—differ fundamentally from those of gallic acid. Although the comparison provides valuable contextual insight, it does not establish therapeutic equivalence, particularly given the disparities in metabolic pathways and timing of administration. This methodological limitation warrants explicit acknowledgment and discussion in the Discussion section. Response 2: We sincerely appreciate this insightful methodological critique. We have revised the Discussion to explicitly state: Notably, NAC and GA operate via distinct pharmacological mechanisms: NAC acts as a rapid-onset GSH precursor, while GA functions as a multi-target polyphenol with prolonged effects. Their comparison serves as a therapeutic benchmark rather than implying mechanistic equivalence (Line 385-388) 3. Lines 285–296 / Figure 8: The Western blot results, while visually suggestive, are not accompanied by quantitative analysis such as densitometry, which is essential to substantiate claims of significant modulation of NF-κB and MAPK protein levels. The absence of such quantitative validation diminishes the strength of the mechanistic interpretations presented. Response 3: We sincerely thank the reviewer for highlighting this critical methodological omission. We have added densitometry methodology in Methods (Line 170-172). Protein band intensity was quantified using Quantity One® software (v4.6.6, Bio-Rad). Target protein expression was normalized to controls and presented as relative expression. 4. Lines 191–203 / Figure 3: TUNEL staining identifies DNA fragmentation, a hallmark of both apoptotic and necrotic cell death. However, the authors attribute the observed TUNEL-positive signals exclusively to apoptosis, which is not accurate within the context of APAP-induced liver injury, where necrotic mechanisms are also prominent. To prevent potential misinterpretation, this point should be clarified and the dual relevance of TUNEL staining acknowledged. Response 4: We sincerely appreciate the reviewer's crucial clarification regarding TUNEL staining interpretation. We have revised text to acknowledge TUNEL's dual relevance (Line 22, 106, 225, 229, 372). 5. Lines 285–303 / Figure 8: Although GA treatment appears to diminish the phosphorylation levels of MAPKs and NF-κB, this observation alone does not conclusively demonstrate functional pathway inhibition. Definitive evidence would require analysis of downstream effectors or the use of pathway-specific reporter assays. Accordingly, the current interpretation should be tempered to more accurately reflect the limitations of the data presented. Response 5: We sincerely appreciate the reviewer's insightful critique regarding pathway validation. We have corrected the expressions (Line 3, 320, 332, 333) Desirable Improvements 1. Lines 309–312 / Figure 9: The involvement of AMPK was explored through the use of Compound C; however, this inhibitor is documented to exert off-target effects that may confound interpretation. It is therefore important that the limitations associated with the specificity of this pharmacological tool be clearly acknowledged and discussed. Response 1: We sincerely appreciate the reviewer's critical insight regarding Compound C's specificity limitations. We have acknowledged and discussed the limitation of NAC (423-425). 2. Line 83: Tissue collection was performed 12 hours after APAP administration, which is a generally acceptable timepoint in this model. However, it would be beneficial to specify whether this interval corresponds to the peak of hepatic injury and maximal inflammatory response, as such temporal alignment is critical for accurately interpreting the observed pathological and molecular outcomes. Response 2: We thank the reviewer for this valuable methodological insight. The 12-hour post-APAP timepoint was strategically selected based on the previous studies. The APAP-induced model of liver injury is a classic model, and we have referred to many previous studies, and some of them are listed in the article. At the same time, our team's previous research (85-86). 3. Lines 130–139: The robustness of the molecular docking analysis could be significantly enhanced by incorporating known ligands for each target protein as positive controls. This would provide a valuable reference framework, allowing for a more meaningful interpretation of GA’s binding affinities in relation to established ligand-target interactions. Response 3: Thanks for you suggestion. In the manuscript, we have shown the affinity between GA and individual proteins (Line 308-309, Table 3) 4. Lines 81–82: The manuscript does not provide a clear rationale for selecting the 50 and 100 mg/kg doses of gallic acid. It would be important to clarify whether these dosing regimens were informed by prior studies on pharmacokinetics, toxicity, or therapeutic efficacy. Including such justification would enhance the translational significance and scientific credibility of the dosing strategy employed. Response 4: We sincerely appreciate the reviewer's essential query regarding GA dosing rationale. Furthermore, preliminary experiments confirmed no observable toxicity at these doses, as assessed through hepatic histopathology and functional biomarkers (Line 85-86). Statistical Review Required? A thorough statistical review is strongly advised to ensure the analytical integrity and validity of the study’s findings. Particular scrutiny should be directed toward Figures 1–9 to confirm the correct implementation of one-way ANOVA and subsequent Tukey’s post-hoc analyses across all experimental groups and outcome measures. Additionally, it is crucial to verify whether foundational statistical assumptions—specifically, data normality and homogeneity of variances—were appropriately assessed and satisfied. Given the considerable number of simultaneous comparisons performed (e.g., cytokine profiles, gene expression data, enzymatic activity), the use of multiple comparison adjustments—such as the Bonferroni correction—should be considered to mitigate the risk of type I statistical errors. Response: We sincerely appreciate the reviewer's meticulous statistical critique. Our analysis did employ Tukey's test following one-way ANOVA, as evidenced by the software output.
